# Simulated terrestrial runoff shifts the metabolic balance of a coastal Mediterranean plankton community towards heterotrophy

Tanguy Soulié[1], Francesca Vidussi[1], Justine Courboulès[1], Marie Heydon[1], Sébastien Mas[2], Florian Voron[2], Carolina Cantoni[3], Fabien Joux[4], Behzad Mostajir[1]

[1]MARBEC (MARine Biodiversity, Exploitation and Conservation), Univ Montpellier, CNRS, Ifremer, IRD, Montpellier, France
[2]MEDIMEER (MEDIterranean Platform for Marine Ecosystems Experimental Research), OSU OREME, CNRS, Univ Montpellier, IRD, INRAE, Sète, France
[3]CNR-ISMAR (Istituto di Scienze Marine), Area Science Park, Basovizza, Ed. Q2, Trieste, Italy
[4]Sorbonne Université, CNRS, Laboratoire d'Océanographie Microbienne (LOMIC), Observatoire Océanologique de Banyuls, Banyuls/Mer, France

Correspondence to: Tanguy Soulié (tanguy.soulie@gmail.com), Behzad Mostajir (behzad.mostajir@umontpellier.fr)

**Abstract.** Climate change is projected to increase the frequency and intensity of extreme rainfall events in the Mediterranean region, increasing runoffs of terrestrial matter into coastal waters. To evaluate the consequences of terrestrial runoff on plankton key processes, an *in situ* mesocosm experiment was conducted for 18 days in the spring of 2021 in the coastal Mediterranean Thau Lagoon. Terrestrial runoff was simulated in replicate mesocosms by adding soil from an adjacent oak forest that had matured in water from the main river tributary of the lagoon. Automated high-frequency monitoring of dissolved oxygen, chlorophyll-a fluorescence, salinity, light, and temperature was combined with manual sampling of organic and inorganic nutrient pools, pH, carbonate chemistry and maximum quantum yield (Fv:Fm) of photosystem II (PSII). High-frequency data were used to estimate gross oxygen primary production (GPP), community respiration (CR), and phytoplankton growth (μ) and loss (L) rates. During the first half of the experiment (d2-d11), the simulated runoff reduced light availability (-52%), chlorophyll-a concentrations (-70%) and phytoplankton growth rates (-53%). However, phytoplankton maintained a certain level of primary production by increasing its photosynthetic efficiency. Meanwhile, the runoff enhanced CR (+53%), shifting the metabolic status (GPP:CR) of the system toward heterotrophy and increasing the partial pressure of carbon dioxide ($pCO_2$), potentially switching the direction of the air-sea $CO_2$ exchange. However, during the second part of the experiment (d11-d17), remineralised nutrients boosted phytoplankton growth (+299%) in the terrestrial runoff treatment, but not its loss rates, leading to phytoplankton biomass accumulation and suggesting a mismatch between phytoplankton and its predators. Our study showed that a simulated terrestrial runoff significantly affected key plankton processes, suggesting that climate change-related increases in runoff frequency and intensity can shift the metabolic balance of Mediterranean coastal lagoons towards heterotrophy.

# 1 Introduction

Climate change is predicted to increase the frequency and intensity of short extreme rainfall events in the Mediterranean region (Alpert *et al.* 2002, Sanchez *et al.* 2004). Consequently, the runoff of terrestrial matter will become more frequent in coastal Mediterranean waters. These runoffs constitute a pulse input of organic and inorganic nutrients into the water column and decrease light penetration (Nunes *et al.* 2009), substantially impacting marine ecosystems, and notably plankton communities (Deininger and Frigstad 2019, Striebel *et al.* 2023).

Plankton is crucial for aquatic ecosystems because it forms the basis of the aquatic food web and plays an important role in multiple biogeochemical cycles, notably that of oxygen (Falkowski *et al.* 2003, Falkowski 2012). Indeed, phytoplankton produces oxygen through its gross primary production (GPP), and all planktonic organisms consume it through aerobic community respiration (CR). Hence, assessing GPP and CR provides a community metabolism index (GPP : CR) and determines the capacity of an aquatic ecosystem to serve as a net producer or consumer of oxygen, and as a sink or source of atmospheric carbon dioxide (Lopez-Urrutia *et al.* 2006). This community metabolism index considerably depends on the fate of phytoplankton, which is itself related to phytoplankton growth (μ) and loss (L) rates. Therefore, assessing μ and L provides a trophic index (μ : L) related to the performance of both phytoplankton and its predators (Soulié *et al.* 2022a).

The consequences of terrestrial runoffs on plankton communities and associated processes remain unclear. The inputs of terrestrial carbon and nutrients have been shown to promote phytoplankton and bacteria in Mediterranean coastal waters (Pecqueur *et al.* 2011, Liess *et al.* 2016), possibly leading to higher GPP and CR. However, this positive effect of nutrient enrichment can be mitigated by light attenuation resulting from the runoff, which can depress phytoplankton photosynthesis, and therefore GPP, as observed in the North Sea, Baltic Sea, and in a North Atlantic bay (Mustaffa *et al.* 2020, Paczkowska *et al.* 2020, Soulié *et al.* 2022b). The contradictory effects of light attenuation and nutrient enrichment induced by terrestrial runoffs on plankton metabolism can change the structure of planktonic communities and, ultimately, their related processes. They can favour bacteria over phytoplankton (Meunier *et al.* 2017, Andersson *et al.* 2018, Courboulès *et al.* 2023), large phytoplankton species at the expense of smaller cells (Deininger *et al.* 2016, Mustaffa *et al.* 2020), and decrease the abundance of protozooplankton (Courboulès *et al.* 2023). Consequently, these shifts can alter plankton processes because the structure and functions of aquatic communities are closely related (Giller *et al.* 2004).

Although the consequences of terrestrial runoffs have been well-studied in freshwater systems, an important knowledge gap exists regarding the impacts of terrestrial runoffs on coastal marine ecosystems (Blanchet *et al.* 2022). In this regard, evaluating the consequences of terrestrial runoffs on plankton communities and processes in ecologically and economically important areas, such as coastal lagoons (Soria *et al.* 2022), enclosed systems that are often subject to inputs from the land, is of fundamental concern. In the present study, we conducted an *in situ* mesocosm experiment in the Mediterranean coastal Thau Lagoon, a shallow productive lagoon which hosts oyster farms and serves as a nursery for several wild fish species (La Jeunesse *et al.* 2015). Moreover, it is naturally subjected to storm-induced terrestrial runoffs (Pecqueur *et al.* 2011, Fouilland *et al.* 2012), notably in fall during the 'Cévenols' events, a meteorological phenomenon characterized by storms and heavy rainfalls

that usually cause flash-flooding ~~in~~ on the Mediterranean coast (Ducrocq *et al*. 2008). Six mesocosms were used with half serving as control mesocosms and in the other half a terrestrial runoff was simulated by adding soil from an adjacent typical Mediterranean oak forest that matured over two weeks in water from the Vène River, the main river tributary of the Thau Lagoon (Plus *et al*. 2006). The responses of all plankton food web compartments in the present experiment have been detailed

by Courboulès *et al*. (2023). In the present study, high-frequency data from automated sensors immersed in the mesocosms were used to estimate GPP, CR, μ and L in each mesocosm, and assess how both the metabolic and trophic indices of the community responded to the simulated runoff. Manual sampling was performed to assess dissolved and particulate materials as well as photosynthetic efficiency and carbonate system parameters. We hypothesized that (1) the metabolic index would be shifted by the runoff towards heterotrophy through light reduction and inputs of organic matter, and that (2) the terrestrial

runoff would affect the trophic index by creating imbalance between phytoplankton and its factors of loss.

## 2 Material and Methods

### 2.1 In situ mesocosm experimental set-up

An *in situ* mesocosm experiment was performed for 18 days in May 2021 in the Thau Lagoon (France) using the facilities of

the MEDIterranean platform for Marine Ecosystems Experimental Research (MEDIMEER, 43°24'53''N, 3°41'16''E). The duration of the experiment was set to 18 days in order to monitor the responses of plankton at medium-term (multiple days to weeks), as interesting dynamics were already reported in control treatments during previous experiments in the Thau Lagoon up to almost 3 weeks after the start of the experiment (Courboulès *et al*. 2021, Soulié *et al*. 2022a). However, the duration of the experiment was limited by COVID-19 pandemics restrictions, preventing from conducting a longer experiment.  The Thau

Lagoon is a shallow coastal lagoon of 75 km$^2$ with a mean depth of 4 m and is located on the French coast of the Northwestern Mediterranean Sea (Derolez *et al*. 2020a). Six mesocosms were deployed in the lagoon. Each mesocosm consisted of a bag, sealed at the bottom, made of nylon-reinforced 200 μm thick vinyl acetate polyethylene film which was 3 m high and 1.2 m wide, resulting in a total volume of 2200 L (Insinööritoimisto Haikonen Ky, Sipoo, Finland). Each mesocosm was equipped with a sediment trap at the bottom. A schematic representation of the mesocosm set-up can be found in Soulié *et al*. (2021)

and in **Supplementary Information**. Each mesocosm was covered with a dome of polyvinyl-chloride to avoid external inputs. On May 3 (d0), all the mesocosms were filled simultaneously using a pump (SXM2/A SG, Flygt) with 2200 L of subsurface lagoon water preliminarily screened through a 1 mm mesh to remove large particles and organisms. The water was pooled in a large container before being distributed simultaneously by gravity to the six mesocosms through parallel pipes. In each mesocosm, the water column was continuously homogenized with a pump (Rule, Model 360) immersed at a depth of 1 m,

resulting in a turn-over rate of approximately 3.5 d$^{-1}$. Observations performed with a microscope indicated that organisms (phytoplankton, zooplankton) did not seem to be damaged by the mixing procedure, even fragile organisms such as ciliates. Three mesocosms served as controls, while in three others maturated soil was added on May 4 (d1) to simulate a terrestrial

runoff event (these mesocosms are hereafter referred to as the "terrestrial runoff" treatment). Throughout the experiment, a total of 510 L was sampled from each mesocosm, representing 23% of the initial volume of the mesocosms. For each treatment, one mesocosm displayed considerable differences in biological, physical, and chemical parameters compared to the two other replicates of the same treatment, most probably because of the malfunctioning of the mixing pumps, and it was therefore removed from the analysis. Data are therefore presented as the mean of the two replicates for each treatment ± the range of observations. Thus, any interpretation of the presented data must take into account the low number of replication and be done cautiously.

## 2.2 Soil extraction, preparation, and maturation

Two weeks before the beginning of the mesocosm experiment, soil was extracted from the Puéchabon state forest, a fully preserved typical Mediterranean oak forest located approximately 30 km north of the Thau Lagoon (43°44'29''N, 3°35'45''E) (Allard *et al.* 2008). The soil was then roughly screened over a 1 cm mesh. On the same day as soil extraction, water was collected from the Vène River, the main tributary of the Thau Lagoon, which is known for its episodic flash floods (Pecqueur *et al.* 2011). Water was screened over a 200 µm mesh to remove large particles and organisms. The soil and river water were then mixed to reach a concentration of 416 g soil $L^{-1}$, which represents natural flash flood events occurring in the lagoon (Fouilland *et al.* 2012). This mixture was then left to mature for two weeks in transparent Nalgene carboys placed in an outdoor pool continuously supplied with natural water from the Thau Lagoon. During the maturation step, each carboy was homogenised and aerated daily. This maturation was performed to mimic the degradation process of the most labile compounds that naturally occurs during their transportation from the soil to coastal waters during natural runoff events (Müller *et al.* 2018). After the manual mesocosms sampling on May 4 (d1), 7 L of the soil solution was added to each of the three "terrestrial runoff" mesocosms, representing a final concentration of 1.3 g soil $L^{-1}$. Further details regarding the choice and description of the soil addition protocol can be found in Courboulès *et al.* (2023).

## 2.3 Acquisition, calibration, and correction of the high-frequency sensor data

In each mesocosm, a set of high-frequency sensors was immersed to a depth of 1 m. Each set consisted of a fluorometer (ECO-FLNTU, Sea-Bird Scientific, United States) for Chl-*a* fluorescence, from which Chl-*a* concentration was derived, an oxygen optode (3835, Aanderaa, Bergen, Norway) for dissolved oxygen (DO) concentration and saturation, an electromagnetic induction conductivity sensor (4319, Aanderaa, Bergen, Norway) for salinity, a spherical underwater quantum sensor (Li-193, Li-Cor, United States) for the incident photosynthetically available radiation (PAR), and three water temperature probes (Thermistore Probe 107, Campbell Scientific, United States) installed at three different depths (0.5, 1 and 1.5 m). Each sensor recorded measurements every minute during the entire experiment. In the results section, the high-frequency data are presented as daily averages. The fluorometers, oxygen optodes, conductivity sensors, and temperature probes were calibrated before and

after the experiment. In addition, Chl-*a* fluorescence and oxygen sensor data were corrected using discrete high-performance liquid chromatography (HPLC) Chl-*a* and Winkler DO measurements, respectively. To do so, three borosilicate bottles (120 mL) were filled with water sampled from each mesocosm using a 5 L Niskin water sampler at a depth of 1 m every other day in the morning. DO was immediately fixed by adding Winkler reagents (Carrit and Carpenter 1966). After at least 6 hr of fixation during which bottles were kept underwater and in the dark in opaque plastic tanks filled with freshwater at room temperature, the DO concentration in each bottle was measured with an automated Winkler titrator (Metrohm 916-Ti-Touch) using a potentiometric titration method. Similarly, a polycarbonate bottle (2 L) was filled with water that was sampled every morning from each mesocosm using a Niskin water sampler at a depth of 1 m. Samples were then immediately filtered under low light conditions using a vacuum pump on glass-fibre filters (Whatman GF/F, 0.7 µm pore size). Filters were then stored at -80 °C until analyses with HPLC (Shimadzu) following the method of Zapata *et al*. (2000). The HPLC was composed of a pump (DGU-405, Shimadzu), an automatic injector (LC-40D, Shimadzu), a Pelletier oven (CTO-40S, Shimadzu), a PDA detector (SPD-M40, Shimadzu), and a fluorimeter detector (RF-20A, Shimadzu). The stationary phase used is a C8 column (C8 Sunfire, Waters) with a C8 column guard (Waters). Details of the calibration procedure can be found in the **Supplementary Information** and in Soulié *et al*. (2023).

### 2.4 Manual mesocosm sampling and monitoring for chemical variables

Each mesocosm was sampled daily using a 5 L Niskin water sampler at a depth of 1 m to monitor dissolved inorganic nutrients (nitrate + nitrite [$NO_2^-$+$NO_3^-$], ammonium [$NH_4^+$], and orthophosphate [$PO_4^{3-}$]), dissolved organic carbon (DOC), particulate organic carbon (POC), and nitrogen (PON) concentrations; and every second day to measure pH and total alkalinity (TA). For dissolved inorganic nutrient analyses, 50 mL sub-samples of mesocosm water were placed in acid-washed polycarbonate bottles. Directly after, these samples were filtered over 0.45 µm filters (Gelman Sciences, United States) and stored in high-density polyethylene tubes at -20 °C until further analyses that were performed within 48 hr. Nitrate, nitrite, and orthophosphate analyses were performed with an automated colorimeter (Skalar Analytical, The Netherlands, Aminot and Kérouel 2007), and ammonium analyses were performed using the fluorometric method (Turner Design, module 7200-067-W, United States, Aminot *et al*. 1997, Holmes *et al*. 1999). For DOC analyses, 30 mL subsamples of mesocosm water were filtered through two pre-combusted (4h, 450 °C) glass-fibre filters (Whatman GF/F), 90 µL of phosphoric acid (85% concentration) was then added and sub-samples were then stored at 4 °C in the dark pending analyses, which were performed by high-temperature catalytic oxidation (HTCO) on a total organic carbon analyser (TOC-L-CSH, Shimadzu). For POC and PON analyses, sub-samples (0.5-1 L) of mesocosm water were filtered over pre-combusted (4h, 450 °C) glass-fibre filters (Whatman GF/F). Filters were then placed in a stove at 60 °C for at least 12h. The POC and PON concentrations were then measured using a CHN analyser (Unicube, Elementar). The samples for pH and TA determinations were collected in 300 mL borosilicate glasses bottles according to standard sampling methods for carbonate chemistry (Dickson *et al*. 2007). Samples for TA determination were

filtered immediately on glass-fibre filters (Whatman GF/F, 0.45 µm pore size), spiked with 50 µL of $HgCl_2$ saturated solution and stored for later analysis. Samples for pH analysis were spiked with $HgCl_2$ and were analysed within 36 hr.

pH was measured spectrophotometrically (LAMBDA 365 UV/Vis, Perkins Elmer), on a "total scale" at 25.0 °C ($pH_{25}$) with m-cresol purple (m-Cp) as an indicator (reproducibility $\pm$ 0.002), and thermostated cells (10 cm optical path, 28 mL), according to Clayton and Byrne (1993) and Dickson *et al.* (2007) with duplicate analysis for control and triplicate for treated mesocosms. TA was measured in the laboratories of CNR-ISMAR in Trieste (duplicate analysis), using an open-cell potentiometric titration with a derivative determination of the end point, according to Hernandez-Ayon *et al.* (1999) (reproducibility $\pm$ 0.1 µmol $kg^{-1}$). Certified reference seawater for carbonate chemistry (provided by Prof A. G. Dickson, Scripps, California) was used for TA analysis and to check the stability of pH analysis during the experiment. After the experiment, the m-Cp used was checked against a purified m-Cp batch (Liu *et al.* 2011), showing a difference < 0.005 pH units. The dissolved inorganic carbon (DIC) concentration, $CO_2$ partial pressure ($pCO_2$), and pH at *in situ* temperature (pH) were calculated using the CO2SYS program (Microsoft Excel version 2.5; Lewis and Wallace 1998, Pierrot *et al.* 2006), using the carbonate constants from Lueker *et al.* (2000) sulphate constants from Dickson (1990), and parameterization of borate from Lee *et al.* (2010).

## 2.5 Estimation of the Daily Light Integral from the high-frequency PAR sensor data

PAR measurements were used to calculate the Daily Light Integral (DLI). This value corresponds to the average quantity of light available for photosynthesis received by a 1 $m^2$ surface over a 24-h period (Soulié *et al.* 2022b). DLI was calculated using **Eq. 1** as follows:

$$DLI = \frac{mean\ PAR \times day\ length \times 3600}{1 \times 10^6}, \quad (\textbf{Equation 1})$$

where DLI is expressed in mol $m^{-2}$ $d^{-1}$, mean PAR between sunrise and sunset in µmol $m^{-2}$ $s^{-1}$, and day length in hr.

## 2.6 Estimation of µ and L from the high-frequency Chl-*a* sensor data

The high-frequency Chl-*a* data were used to estimate phytoplankton growth (µ) and loss (L) rates following a method detailed by Soulié *et al.* (2022a). First, the high-frequency Chl-*a* data were corrected for non-photochemical quenching as detailed in **Supplementary Information**. Then, each Chl-*a* cycle was separated into an "increasing period" and a "decreasing period". The "increasing period" started at sunrise until the maximum Chl-*a* fluorescence was reached, generally a few minutes to a few hours after sunset. The "decreasing period" started from this maximum until the next sunrise. For each period, an exponential fit was applied to the Chl-*a* data, and L was estimated form the decreasing period. Then, µ was estimated from the increasing period. The detailed calculations are presented in the **Supplementary Information**.

## 2.7 Estimation of GPP and CR from the high-frequency DO sensor data

DO data were used to estimate daily GPP, CR during the day (CRdaytime) and the night (CRnight), and daily CR following the method detailed by Soulié *et al.* (2021). This method is derived from the free-water diel oxygen technique (Staehr *et al.* 2010), and was specially developed for mesocosm experiments and to consider variability in both the coupling between day-night and DO cycles and in the respiration occurring during the day and at night. Briefly, each DO cycle was separated into a "positive instantaneous net community production period" (during which DO increases) and a "negative instantaneous net community production period" (during which DO decreases). For each period, the DO was smoothed using a 5-point sigmoidal model. These smoothed data were then used to estimate oxygen metabolic parameters in two major steps. First, the oxygen exchange term between water and the atmosphere was calculated, considering its dependence on temperature and salinity. Then, instantaneous and daily metabolic parameters were estimated. A precise description of the method is provided by Soulié *et al.* (2021) and the **Supplementary Information**.

## 2.8 Maximum photosystem II quantum yield measurements

Phytoplankton photosynthetic performance was estimated based on the fluorescence of the photosystem II (PSII). Subsamples of 1.5 mL from the Niskin water sampler were collected daily and analysed using a portable Pulse Amplitude Modulation fluorometer (Aquapen C AP 110 C, Photon System Instruments, Czech Republic). The maximum quantum yield of photosynthesis ($F_v : F_m$) was measured after a 30-min acclimation period in the dark to ensure that all photosystem-II reactional centres were open. The measurement was done using the 'OJIP' protocol and an excitation wavelength of 450 nm (Strasser *et al.* 2000).

## 2.9 Heterotrophic bacterial abundance measurements

Heterotrophic bacterial abundance was assessed daily using flow cytometry. For this purpose, 1.5 mL samples were collected from the Niskin water sampler and fixed using glutaraldehyde (Grade I, Sigma; 4% final dilution), and then frozen into liquid nitrogen before being maintained at -80°C until further analyses. The samples were stained with SYBR Green I (S7563, Invitrogen; 0.25% final dilution) (Marie *et al.* 1997). Analyses were performed using a FACSCanto2 flow cytometer (Becton-Dickinson; set at low speed for 3 min), and internal cell size standards (cytometry fluorescent beads, Polysciences Inc.) of 1 and 2 µm diameter were added to each run. Bacterial populations were identified and counted via stained green fluorescence (530/30 nm) and relative side scatter (Courboulès *et al.* 2021, 2023).

## 2.10 Statistical analyses

To test the difference between the control and terrestrial runoff treatments, we performed Repeated-Measures Analyses of Variances (RM-ANOVA) with the treatment as a fixed factor and time as a random factor (*nlme* package, R software) over the entire experiment (after the addition of soil, d2-d18) and over shorter periods to assess specific trends. Data from d1 were
225 not included in the statistical analyses as sampling was performed before adding the soil, simulating the terrestrial runoff, in the runoff mesocosms. Statistical significance was set at $p < 0.05$. Before performing the RM-ANOVAs, the assumptions of homoscedasticity and normality were checked using the Levene and Shapiro-Wilk tests, respectively. When these assumptions were not met even after transforming the data (log- or square-root transformation), a non-parametric Kruskal-Wallis test was performed instead of RM-ANOVA. The non-parametric Spearman's correlation coefficient was used to assess significant ($p$
$< 0.05$) relationships between the Logarithm Response Ratio (LRR) of the variables. All data management and statistical analyses were performed using the R software (version 4.0.1).

## 3 Results

### 3.1 Effects of the terrestrial runoff treatment on physical and chemical conditions

In the control treatment, the water temperature varied from $16.68 \pm 0.16$ °C to $17.95 \pm 0.65$ °C (**Fig. 1a**), and was not significantly different in the terrestrial runoff treatment compared to the control (**Table 1**). The salinity was on average $38.42 \pm 0.11$ in the control treatment, increasing almost continuously throughout the experiment (**Fig. 1b**). In the terrestrial runoff treatment, the salinity was significantly reduced by 0.7% (**Table 1**). Similarly, the DLI was, on average, $18.65 \pm 1.45$ mol m$^{-2}$ d$^{-1}$ in the control treatment (**Fig. 1c**). The terrestrial runoff drastically decreased it, by 76% on d2 and by, on average, 43%
over the entire experiment. This negative effect was stronger during the first half of the experiment (52% from d2 to d11), and was attenuated during the second half of the experiment (27% from d12 to d18) (**Table 1**). In the control treatment, pH varied between $8.10 \pm 0.05$ and $8.19 \pm 0.01$ (**Fig. 1d**), decreasing from d1 to d10 before stabilisation until the end of the experiment. In the runoff treatment, it was significantly reduced by on average 0.03 units ($8.06 \pm 0.01$ to $8.19 \pm 0.01$ in the runoff treatment) (**Table 1**). In addition, $pCO_2$ ranged from $292.49 \pm 0.45$ to $368.27 \pm 43.97$ µatm in the control treatment (**Fig. 1e**). In the runoff
treatment, it was significantly higher by 9% compared to the control, despite returning to the control level by the end of the experiment (**Table 1**). DIC concentrations ranged from $2184.04 \pm 14.89$ to $2230.44 \pm 0.76$ µmol kg$^{-1}$ (**Fig. 1f**). They were significantly higher by 1% in the runoff treatment than in the control, with the highest difference between treatments on d2 (3%) (**Table 1**). DOC concentrations were on average $1.70 \pm 0.10$ mg L$^{-1}$ in the control treatment (**Fig. 1g**). In the terrestrial runoff treatment, DOC concentrations were not immediately enhanced after the addition of soil, reaching higher concentrations
than in the control only in the middle and end of the experiment. However, no significant differences were observed between the treatments (**Table 1**). POC + PON concentrations displayed similar dynamics over time. POC concentrations ranged from

0.26 ± 0.01 to 0.55 ± 0.09 mg L$^{-1}$ (**Fig. 1h**), whereas PON concentrations ranged from 0.04 ± 0.01 to 0.07 ± 0.01 mg L$^{-1}$ (**Fig. 1i**). They were both significantly enhanced by 32-50% by the terrestrial runoff at the beginning of the experiment (d2 to d12), then decreased to the level of the control (**Table 1**). The concentrations of dissolved inorganic nutrients exhibited different trends. Nitrate + nitrite concentrations ranged from 0.29 ± 0.03 to 0.50 ± 0.01 µM in the control treatment, and were not significantly affected by the terrestrial runoff (**Fig. 1j**, **Table 1**). Conversely, while ammonium concentrations remained rather constant in the control treatment during the experiment, ranging from 0.02 ± 0.01 to 0.12 ± 0.07 µM, they increased significantly in the terrestrial runoff treatment, reaching 0.96 ± 0.04 µM on d10, before decreasing to the control level on d16 (**Fig. 1k**, **Table 1**). Orthophosphate concentrations ranged from 0.03 ± 0.01 to 0.07 ± 0.01 µM in the control treatment, with peaks at the beginning and the end of the experiment (**Fig. 1l**). They were significantly higher in the terrestrial runoff treatment, but only in the middle of the experiment (63% from d10 to d13) (**Table 1**).

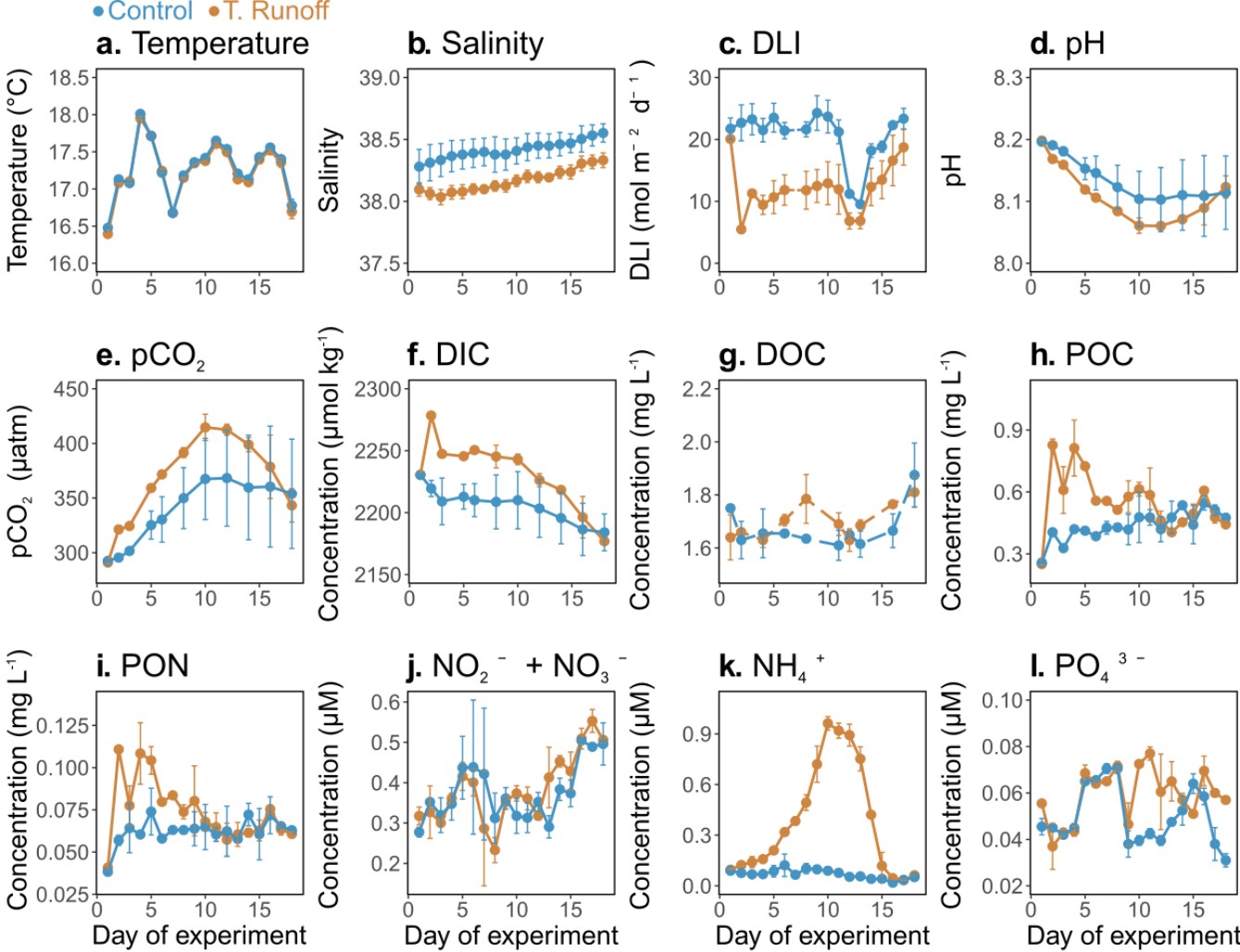

**Figure 1. Daily average temperature (a), salinity (b), daily light integral (DLI, c), pH (d), pCO₂ (e), dissolved inorganic carbon concentrations (DIC, f), dissolved organic carbon concentrations (DOC, g), particulate organic carbon concentrations (POC, h), particulate organic nitrogen concentrations (PON, i), nitrate + nitrite concentrations (NO₂⁻+NO₃⁻, j), ammonium concentrations (NH₄⁺, k), and orthophosphate concentrations (PO₄³⁻, l) in the control (blue) and terrestrial runoff (gold) treatments. Error bars represent the range of the observations (min and max values).**

### 3.2 Effects of the terrestrial runoff treatment on bacterial abundances

In the control treatment, bacterial abundances ranged from $0.8 \times 10^6 \pm 0.3 \times 10^6$ to $1.9 \times 10^6 \pm 0.8 \times 10^6$ cells mL$^{-1}$ (**Fig. 2**). They were significantly higher in the terrestrial runoff treatment from d2 to d8 (59%) and from d15 to d18 (51%), whereas they were significantly lower in the middle of the experiment (-47% from d9 to d14) (**Table 1**). As a consequence, no significant differences were observed when considering the entire duration of the experiment.

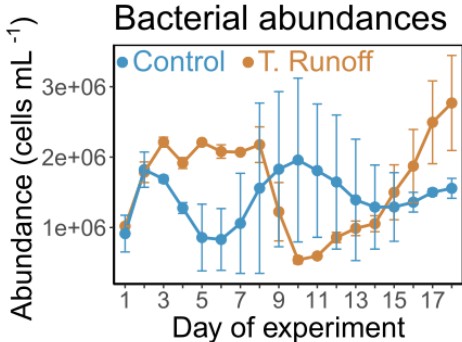

**Figure 2. Daily average bacterial abundances in the control (blue) and terrestrial runoff (gold) treatments. Error bars represent the range of the observations (min and max values).**

### 3.3 Effects of the terrestrial runoff treatment on phytoplankton: Chl-a, growth and loss rates

In the control treatment, the Chl-*a* concentrations ranged from $0.83 \pm 0.30$ µg L$^{-1}$ to $1.91 \pm 0.45$ µg L$^{-1}$ (**Fig. 3a**). They remained rather constant during the first half of the experiment, before increasing from d11 to d13, and then decreasing until the end of the experiment. In the terrestrial runoff treatment, they were significantly lower than in the control, particularly during the first part of the experiment (-70% from d2 to d11) (**Table 1**). However, at the end of the experiment, they increased rapidly from d11 to d15, even surpassing the control level.

In the control, µ ranged from $0.06 \pm 0.04$ d$^{-1}$ to $0.64 \pm 0.06$ d$^{-1}$, peaking on d4, d7 and d12 (**Fig. 3b**). In the terrestrial runoff treatment, it was significantly lower than in the control by an average of 53% from d2 to d10 (**Table 1**). However, it increased drastically during the second half of the experiment, and was significantly almost three times higher than in the control from d12 to d17. L varied from $0.19 \pm 0.12$ d$^{-1}$ to $0.95 \pm 0.05$ d$^{-1}$ in the control treatment, and was rather constant (**Fig. 3c**). In the

terrestrial runoff treatment, it was significantly lower than in the control by an average of 32% throughout the experiment, and by 60% from d3 to d14 (**Table 1**). However, it was higher than in the control from d1̶2 to d3, and came back to the control level at the end of the experiment. As a consequence of the generally higher L than μ in the control, the μ:L ratio was below 1 on 13 out of the 16 days (**Fig. 3d**). It ranged from $0.10 \pm 0.06$ to $2.76 \pm 2.26$. The terrestrial runoff significantly increased the μ:L ratio by an average of 305% over the entire experiment (**Table 1**). The greatest difference between treatments was found on d13, when the ratio was almost 11 times higher in the terrestrial runoff than in the control treatment.

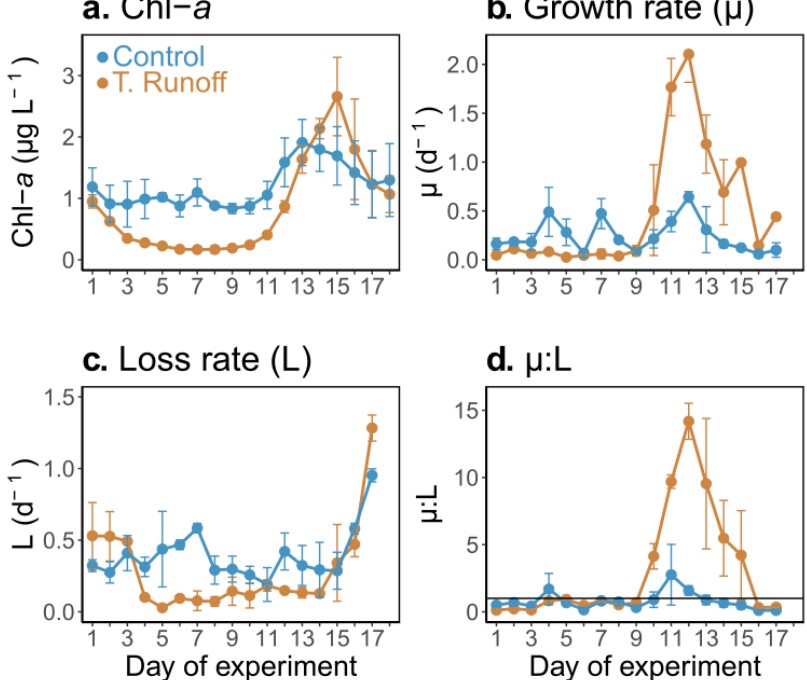

**Figure 3. Daily average chlorophyll-*a* (Chl-*a*, a), phytoplankton growth rate (μ, b), phytoplankton loss rate (L, c), and growth : loss ratio (μ:L, d) in the control (blue) and terrestrial runoff (gold) treatments. Error bars represent the range of the observations (min and max values). Note that μ and L could not be estimated on d18 owing to the lack of a complete fluorescence cycle.**

### 3.4 Effects of the terrestrial runoff treatment on primary production, respiration, and photosynthetic efficiency

In the control treatment, GPP ranged from $0.26 \pm 0.02$ to $0.78 \pm 0.03$ $gO_2$ $m^{-3}$ $d^{-1}$ (**Fig. 4a**). After decreasing from d1 to d2, it increased until it reached its maximum on d7, and then decreased almost continuously until the end of the experiment. In the terrestrial runoff treatment, it increased significantly by an average of 37% at the middle of the experiment, from d9 to d14 (**Table 1**). When the GPP was normalised by the daily Chl-*a* concentration, it was significantly higher in the terrestrial runoff treatment than in the control by an average of 312% throughout the experiment (**Fig. 4b**).

In the control treatment, CR ranged from $0.18 \pm 0.01$ to $0.67 \pm 0.02$ $gO_2$ $m^{-3}$ $d^{-1}$, and it showed a similar dynamic as GPP (**Fig. 4c**). It was significantly enhanced in the terrestrial runoff treatment by an average of 46% over the entire experiment (**Table 1**).

The GPP : CR ratio ranged from $0.94 \pm 0.12$ to $1.69 \pm 0.19$ in the control treatment, and it was higher than 1 on 16 out of 17 days (**Fig. 4d**). In the terrestrial runoff treatment, it decreased significantly by an average of 32% during the first half of the experiment (d2-d10), before increasing and reaching the control level during the second half of the experiment (**Table 1**). Consequently, it was higher than 1 only on 10 out of the 17 days.

In the control, the maximum PSII quantum yield, an indicator of the maximum potential photosynthetic capacity, ranged from
$0.24 \pm 0.01$ to $0.55 \pm 0.04$ (**Fig. 4e**). It was not significantly different between the treatments over the entire experiment; however, it increased significantly by 43% in the terrestrial runoff treatment from d8 to d11 (**Table 1**).

When the CR was normalised by the daily Chl-*a* concentration, it was significantly higher in the terrestrial runoff treatment than in the control by an average of 420% throughout the experiment (**Fig. 4f**).

Finally, when CR was normalised by total bacterial abundance (**Fig. 4g**), it was not significantly different between treatments
apart from d9 to d14 when it was significantly higher by an average of 154% in the terrestrial runoff treatment than in the control (**Table 1**).

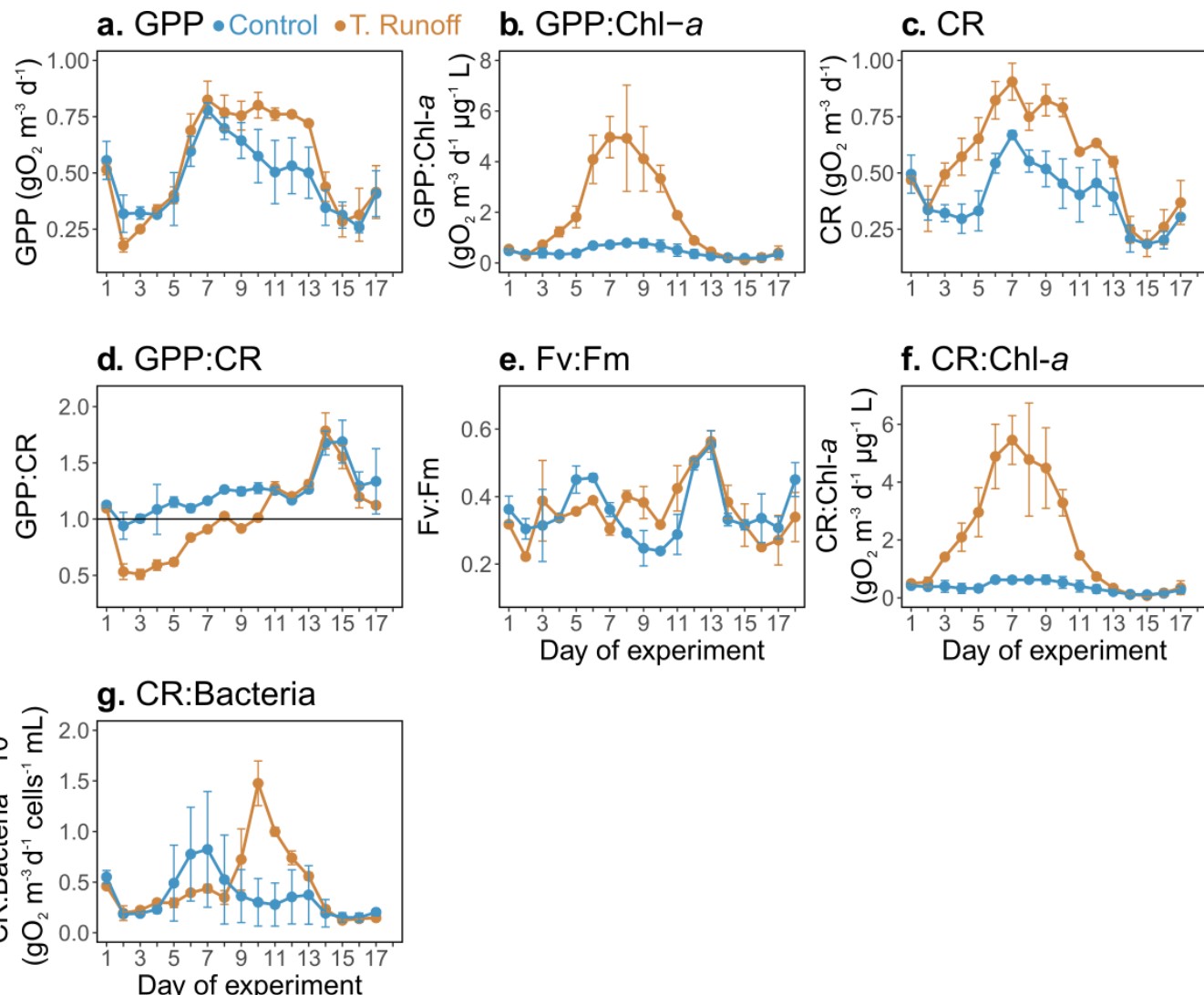

**Figure 4.** Daily average gross primary production (GPP, a), GPP normalised by chlorophyll-*a* (GPP:Chl-*a*, b), community respiration (CR, c), GPP : CR ratio (d), maximum quantum yield ($F_v : F_m$) of photosystem II (PSII) (e), CR normalised by chlorophyll-a (CR:Chl-a, f), and CR normalised by bacterial abundance (CR:Bacteria, g) in the control (blue) and terrestrial runoff (gold) treatments. Error bars represent the range of the observations (min and max values). Note that GPP and CR could not be estimated on d18 owing to the lack of a complete oxygen cycle.

**Table 1.** Summary table of the statistical comparison and the % relative change between the terrestrial runoff and the control treatments. The significance level was set to 0.05 and significant P-values, as well as their corresponding relative change, were highlighted in bold. When a RM-ANOVA was performed, its F value was given in brackets, and when a Kruskal-Wallis was performed instead, "KW" was indicated.

| Parameter | Period | P-value | % difference |
|---|---|---|---|
| Temperature | 2-18 | 0.64 (KW) | -0.2 |
| Salinity | 2-18 | **$< 1\times10^{-4}$ ($F_{1,16}$=1035)** | **-0.7** |

| | | | |
|---|---|---|---|
| | 2-18 | $1.4\times10^{-3}$ (KW) | **-43.3** |
| DLI | 2-11 | $9.1\times10^{-4}$ (KW) | **-51.6** |
| | 12-18 | $1.4\times10^{-3}$ ($F_{1,6}$=32.9) | **-27.3** |
| pH | 2-18 | $3\times10^{-4}$ ($F_{1,9}$=32.4) | **-0.4** |
| $pCO_2$ | 2-18 | $4\times10^{-4}$ ($F_{1,9}$=30.5) | **8.9** |
| DIC | 2-18 | $7\times10^{-4}$ ($F_{1,9}$=25.3) | **1.3** |
| DOC | 2-18 | 0.27 (KW) | 0.4 |
| | 2-18 | $2.2\times10^{-6}$ ($F_{1,16}$=11.5) | **27.8** |
| POC | 2-12 | $1.1\times10^{-8}$ ($F_{1,10}$=27.9) | **49.3** |
| | 12-18 | 0.621 (KW) | -2.0 |
| | 2-18 | **0.001** (KW) | **18.8** |
| PON | 2-12 | $1.6\times10^{-5}$ ($F_{1,10}$=12.9) | **32.3** |
| | 12-18 | 0.474 ($F_{1,6}$=0.7) | -2.7 |
| $NO_2^- + NO_3^-$ | 2-18 | 0.75 ($F_{1,16}$=0.1) | 1.3 |
| $NH_4^+$ | 2-18 | $3.2\times10^{-4}$ (KW) | **486.5** |
| $PO_4^{3-}$ | 2-18 | **0.02** ($F_{1,16}$=6.8) | **18.0** |
| | 10-13 | $8.4\times10^{-3}$ ($F_{1,3}$=38.7) | **62.7** |
| | 2-18 | 0.183 (KW) | 15.0 |
| Bacterial abundances | 2-8 | $1.3\times10^{-4}$ ($F_{1,6}$=37.2) | **59.0** |
| | 9-14 | $2.5\times10^{-3}$ ($F_{1,5}$=24.2) | **-47.0** |
| | 15-18 | $6.7\times10^{-3}$ ($F_{1,3}$=22.7) | **51.0** |
| | 2-18 | $1.2\times10^{-3}$ ($F_{1,17}$=14.9) | **-30.2** |
| Chl-*a* | 2-11 | $1.6\times10^{-4}$ (KW) | **-70.2** |
| | 12-18 | 0.89 ($F_{1,8}$=0.02) | 4.2 |
| | 2-17 | 0.86 ($F_{1,15}$=0.03) | 110.6 |
| Growth rate (μ) | 2-11 | **0.02** ($F_{1,8}$=7.5) | **-52.8** |
| | 12-17 | $3.0\times10^{-4}$ ($F_{1,5}$=77.9) | **298.7** |
| Loss rate (L) | 2-17 | $4.7\times10^{-3}$ ($F_{1,15}$=11) | **-32.1** |
| | 3-14 | $6\times10^{-4}$ ($F_{1,11}$=22.3) | **-60.0** |
| μ : L ratio | 2-17 | **0.02** ($F_{1,15}$=7.3) | **305.4** |
| | 11-18 | $1\times10^{-4}$ ($F_{1,5}$=115) | **550.3** |
| | 2-17 | 0.37 (KW) | 16.1 |
| GPP | 9-14 | $1.1\times10^{-3}$ ($F_{1,5}$=44.7) | **36.6** |
| | 12-17 | 0.08 ($F_{1,15}$=4.8) | 24.5 |
| GPP : Chl-*a* | 2-17 | **0.02** (KW) | **312.1** |
| CR | 2-17 | $<1\times10^{-4}$ ($F_{1,15}$=38.4) | **45.7** |
| | 2-11 | $2\times10^{-4}$ ($F_{1,10}$=32.7) | **52.5** |
| GPP : CR | 2-17 | $7\times10^{-4}$ ($F_{1,15}$=18.4) | **-17.6** |
| | 2-10 | $<1\times10^{-4}$ ($F_{1,15}$=82.5) | **-32** |
| $F_v : F_m$ | 2-17 | 0.94 ($F_{1,17}$=0.01) | 0.4 |
| | 8-11 | $3.7\times10^{-3}$ ($F_{1,5}$=68.7) | **43.0** |
| CR : Chl-*a* | 2-17 | $7.5\times10^{-4}$ (KW) | **419.9** |
| CR : Bacteria | 2-17 | 0.16 (KW) | 26.9 |
| | 9-14 | $2\times10^{-4}$ ($F_{1,5}$=22.1) | **154.2** |


**3.5 Correlation matrix between the responses of phytoplankton processes, community metabolism, and environmental variables**

To assess the relationships between the effects of the terrestrial runoff on various variables, Spearman's correlations were calculated between the LRR of phytoplankton processes, community metabolism and environmental variables. All significant correlations are shown in the matrix (**Fig. 5**). GPP was positively correlated with $NH_4^+$ and $PO_4^{3-}$ concentrations, and negatively correlated with bacteria abundance and POC+PON concentrations. CR was positively correlated with $pCO_2$ and POC+PON concentrations, while being negatively correlated with μ, Chl-*a*, salinity, DLI and pH. In addition, μ was positively correlated to L, Chl-*a*, salinity, DLI, $NO_2^-+NO_3^-$, and negatively to bacterial abundances and POC+PON concentrations. Similarly, L was positively correlated with Chl-*a* and salinity, and negatively correlated with $pCO_2$. In addition, Chl-*a* was positively correlated with salinity, DLI and $NO_2^-+NO_3^-$, and negatively correlated with POC+PON, while bacterial abundances were positively correlated with DOC, and negatively correlated with $NO_2^-+NO_3^-$, $NH_4^+$ and $PO_4^{3-}$. Among environmental variables, it should be noted that DLI and POC+PON concentrations were negatively correlated, and $NH_4^+$ and $PO_4^{3-}$ were positively correlated.

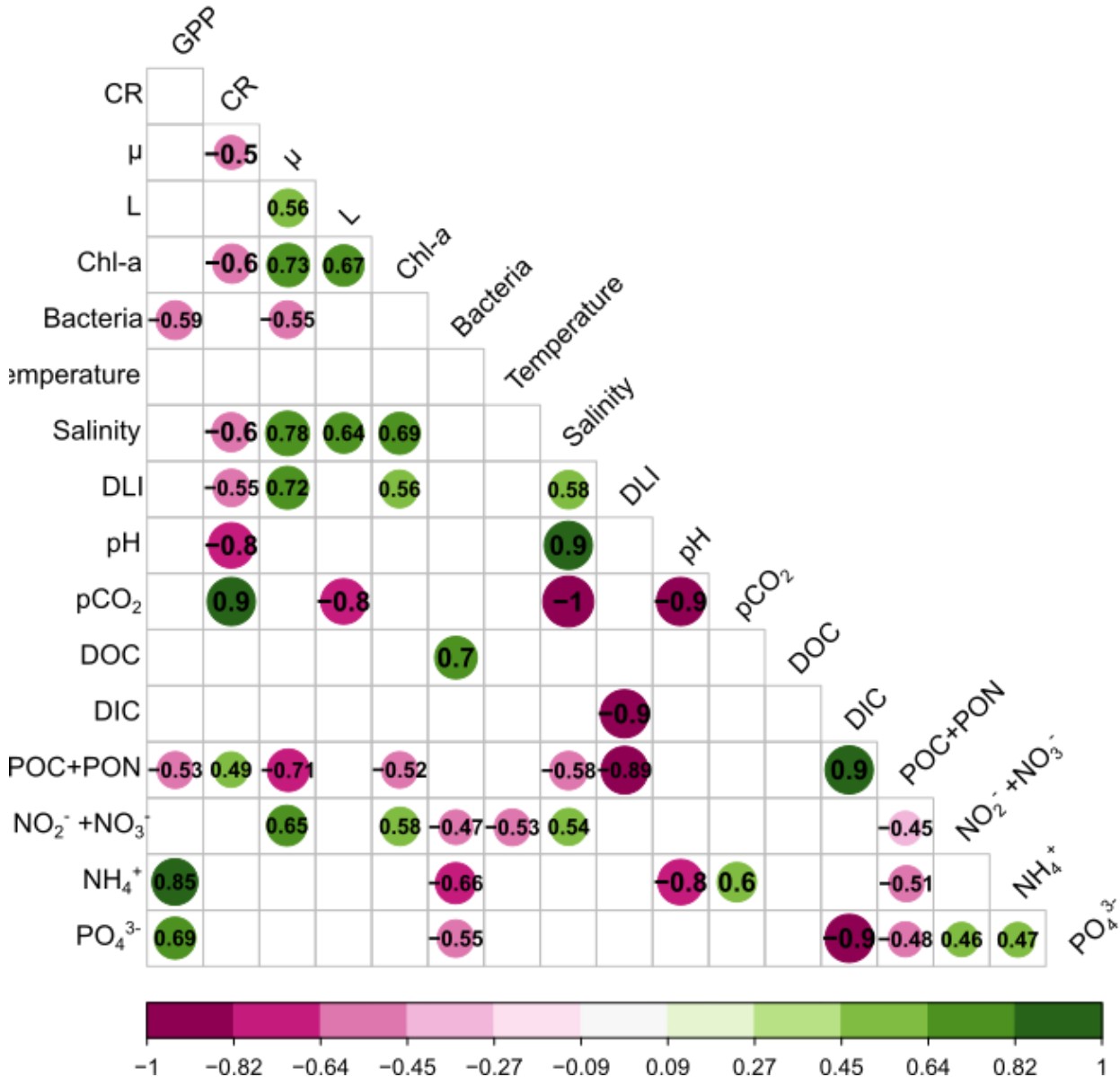

**Figure 5. Correlation matrix based on Spearman's correlations between the log response ratio (LRR) of phytoplankton processes, community metabolism, and environmental variables. Only significant ($p < 0.05$) correlations are shown in the matrix. Green illustrates positive correlations and purple negative correlations. (GPP: Gross Primary Production, CR: Respiration, μ: Growth rate, L: Loss rate, Chl-*a*: Chlorophyll-*a*, Bacteria: bacterial abundance, DLI: Daily Light Integral, pCO₂: partial pressure of CO₂, DOC: Dissolved Organic Carbon, DIC: Dissolved Inorganic Carbon, POC + PON: Particulate Organic Carbon + Nitrogen, NO₂⁻**
**+NO₃⁻: nitrite + nitrate concentrations, NH₄⁺: ammonium concentration, PO₄³⁻: orthophosphate concentration).**

## 4 Discussion

### 4.1 The terrestrial runoff depressed phytoplankton processes and shifted the metabolic balance of the system towards heterotrophy during the first half of the experiment

The present study aimed to evaluate the effects of a simulated terrestrial runoff on key plankton processes in a coastal Mediterranean lagoon. During the first half of the experiment (d2-d11), the simulated terrestrial runoff strongly decreased available light (-52%), consequently depressing phytoplankton biomass (-70%) and growth rate (-53%), as highlighted by the strong positive correlations between light availability, Chl-*a* and phytoplankton growth. The phytoplankton community investigated in the present study was typical of the Thau Lagoon in spring (Trombetta *et al.* 2019), mainly composed of

diatoms, cryptophytes, and small nano- and picophytoplankton (Courboulès *et al.* 2023). The negative effect of light limitation induced by the runoff on phytoplankton biomass is consistent with a mesocosm experiment performed in the Baltic Sea where terrestrial organic matter addition reduced phytoplankton biomass through light attenuation (Mustaffa *et al.* 2020) and, generally, with a meta-analysis conducted on 108 studies reporting an average 23% reduction in photoautotroph biomass in response to experimentally reduced light across various freshwater and coastal ecosystems (Striebel *et al.* 2023). However, in

the Thau Lagoon, a previous experiment reported a positive effect of soil addition, simulating a terrestrial runoff, on phytoplankton (Deininger *et al.* 2016). Nevertheless, the sinking of the added soil during the experiment performed by Deininger *et al.* (2016), despite the use of a mixing pump to limit sedimentation, might have rapidly lessen light attenuation, possibly releasing phytoplankton from the negative effect of light limitation. In addition, the experiment was conducted in late spring / early summer, when light is oversaturating (Trombetta *et al.* 2019), whereas our experiment was performed in spring,

when light could be limiting for phytoplankton metabolism. Finally, Deininger *et al.* (2016) used a resin in their soil extraction procedure, yielding higher inorganic and organic nutrient concentrations in their extract compared to the protocol performed in the present study but being farther from natural terrestrial runoffs (Scharnweber *et al.* 2021). In the present experiment, the maturation step aimed at mimicking processes naturally occurring during the transportation of soil to coastal waters during terrestrial runoffs, such as the degradation of the most labile organic compounds (Müller *et al.* 2018). The 14 d maturation

period can be considered as a long residence time in river water, regarding the fact that flash floods in the Mediterranean region are usually faster. Therefore, it can be supposed that the terrestrial matter added in the present study contained lower levels of labile organic compounds than what can be found during flash floods. This emphasises the need for extreme caution when comparing experimental studies investigating terrestrial runoff effects because protocols are often different from one study to another.

In the present study, the lower phytoplankton biomass and growth rate in the runoff treatment were coupled with an overall decrease in phytoplankton loss rate from d3 until d14 (-60%). Phytoplankton loss could be caused by multiple factors that occur concomitantly, including: grazing by predators, viral lysis, sedimentation and natural death (Landry and Hassett 1982, Brussaard 2004). As the terrestrial runoff induced a negative effect on phytoplankton biomass during the first half of the experiment, it may have led to lower prey availability for its predators, resulting in a lower phytoplankton loss rate. This is

supported by the negative effect of the simulated runoff on protozooplankton abundances reported in the present experiment (Courboulès *et al.* 2023), which may be due to both lower phytoplankton abundance and higher grazing pressure from metazooplankton. Finally, the lower phytoplankton loss rate suggests that terrestrial runoffs could have important consequences for the entire plankton food web of coastal Mediterranean waters by disrupting phytoplankton loss processes, including grazing which the first link in the herbivorous food web (Legendre and Rassoulzadegan 1995, Mostajir *et al.* 2015).

In contrast to phytoplankton biomass and growth, the gross primary production returned quickly to the control level (d4), and was even enhanced by the terrestrial runoff after a few days. This result was unexpected considering that oxygen production strongly depends on light, which was reduced by the runoff. However, we showed that the primary production to Chl-*a* ratio increased by more than three times in the runoff treatment, suggesting a strong enhancement of the phytoplankton photosynthetic efficiency to cope with lower light availability. Supporting this, the maximum PSII quantum yield, an indicator

of the maximum potential photosynthetic activity (Strasser *et al.* 2000), increased significantly in the middle of the experiment in the terrestrial runoff treatment, further suggesting an increase in photosynthetic efficiency under light attenuation induced by the runoff. Moreover, this mismatch between oxygen production and carbon fixation, which has already been reported in a mesocosm experiment in Antarctic coastal waters (Deppeler *et al.* 2018), might be explained by the fact that photosynthetic carbon fixation is a two-stage process. The first is the conversion of light to energy in the chloroplast which produces oxygen

as a by-product, and the second is the use of the produced energy to convert carbon dioxide into sugars through the Calvin cycle with the RuBisCO enzyme. Under stress conditions, the energy produced can also be used in alternative pathways other than carbon dioxide conversion, mainly respiration and photoacclimation (Behrenfeld *et al.* 2004, Halsey *et al.* 2010). Hence, we hypothesised that in the runoff treatment, a significant part of the energy produced by photosynthesis was not converted to growth, but was used instead in alternative pathways, explaining the observed mismatch between oxygen production and

phytoplankton biomass. An alternative hypothesis is that the high quantity of particulate matter added through the simulated runoff induced a strong sedimentation of a part of the phytoplankton community toward the bottom of the mesocosm enclosures (Kiorboe *et al.* 1990). This sedimentation could have partly contributed to the mismatch between GPP and Chl-*a*, as sedimented phytoplankton could have continued to produce oxygen, while being undetected by both manual and sensor monitoring of Chl-*a*. Such sedimentation has already been suggested after heavy loadings of terrestrial matter during a natural flash flood event

in Thau Lagoon, during which most of the microbial production may have been exported through sedimentation (Fouilland *et al*. 2012). Nonetheless, it should be noted that the samples of sedimented material in the sediment traps are not fully analysed yet, thus preventing to characterize the role of sedimentation in the responses of GPP and Chl-*a* with certainty.

Simultaneously, community respiration was strongly enhanced (+53%) by the simulated terrestrial runoff. In marine waters, planktonic bacterial respiration is generally assumed to represent a major part of community respiration (Robinson 2008). In

the present study, bacterial abundance was significantly enhanced by the runoff during the first part of the experiment (d2-d9), which is congruent with the higher respiration at that time. This suggests that higher bacterial abundances are certainly responsible for the higher R reported in the runoff treatment during the first part of the experiment. However, bacterial abundances then significantly decreased during the middle of the experiment (d9-d14) in the runoff treatment, while respiration

remained significantly higher than in the control treatment, resulting in a positive response of R normalised by bacterial abundance at this time of the experiment. This suggests that respiration was mostly not sustained by bacteria at that time of the experiment, but by other biological compartments instead. Because Chl-$a$ was still strongly depressed by the runoff during this period of the experiment, resulting in extremely high CR:Chl-$a$ rates, the hypothesis of an increase in phytoplankton respiration is not plausible. An increase in zooplankton respiration might instead explain the positive effect on community respiration, as the abundance of some groups of metazooplankton was significantly enhanced by the runoff treatment (Courboulès $et$ $al.$ 2023), and the concomitant increase in $PO_4^{3-}$ suggests a strong phosphorus excretion from zooplankton (Andersen $et$ $al.$ 1986, Vadstein $et$ $al.$ 1995).

As a consequence of the faster and greater increase in respiration compared to that in gross primary production, the terrestrial runoff resulted in a decrease in the production to respiration ratio and a shift toward heterotrophy of the metabolic index of the planktonic system during the first half of the experiment, as similarly reported after simulating a terrestrial runoff in a tropical reservoir (Trinh $et$ $al.$ 2016). Concomitantly, $pCO_2$ was significantly higher in the terrestrial runoff treatment, certainly because of the higher respiration as the responses of both variables were strongly correlated. These results are consistent with a study of 15 Swedish lakes that reported higher respiration leading to switches towards a heterotrophic metabolic index and increased $pCO_2$ in response to increased terrestrial carbon runoffs (Ask $et$ $al.$ 2012). Therefore, the present experiment shows, for the first time to our knowledge in Mediterranean coastal lagoons, that terrestrial runoffs could potentially shift coastal Mediterranean lagoons, such as the Thau Lagoon, from being net oxygen producer to net oxygen sink in spring. Therefore, the respiration-driven gain in $CO_2$ can temporarily change the magnitude and direction of the air-sea $CO_2$ exchange, potentially switching the ecosystem from a $CO_2$ sink to a $CO_2$ source for the atmosphere.

**4.2 Enhanced nutrient availabilities boosted phytoplankton processes during the second half of the experiment**

During the second half of the experiment (d12-18), the phytoplankton biomass and processes increased in the terrestrial runoff treatment, in contrast to what occurred during the first half of the experiment. This might be explained by the higher dissolved inorganic nutrient availability in the runoff treatment, as both the $NH_4^+$ and $PO_4^{3-}$ concentrations were significantly higher in the terrestrial runoff treatment than in the control in the middle of the experiment, before being consumed and returning to the control level. The higher $NH_4^+$ concentrations possibly resulted from bacterial remineralization, as $NH_4^+$ is mostly produced by bacterial remineralisation of organic matter in coastal waters (Nixon 1981, Glibert 1982). In contrast, the higher $PO_4^{3-}$ availability could be linked to grazing on bacteria, as grazers feeding upon bacteria generally show high phosphorus excretion rates (Andersen $et$ $al.$ 1986).

Enhanced nutrient availability may have fuelled phytoplankton growth to such an extent that the positive effect of nutrient availability surpassed the negative effect of light attenuation. This result suggests a trade-off mechanism between light and nutrient availability, whereby phytoplankton metabolism is enhanced or depressed depending on the extent of nutrient enrichment compared to the light attenuation associated with terrestrial runoffs. This mechanism has already been reported for

northern lakes (Klug 2002, Isles *et al.* 2021) and even during mesocosm experiments evaluating the addition of dissolved organic matter into coastal waters of various regions (Deininger *et al.* 2016, Traving *et al.* 2017, Andersson *et al.* 2023). The present study provides additional support for this mechanism in Mediterranean coastal waters, and highlights the importance of considering it when modelling their response to terrestrial runoffs.

As mentioned earlier, Chl-*a* strongly increased during the second part of the experiment in the runoff treatment. This positive response was mainly due to an increase in the abundance of diatoms, mainly *Chaetoceros sp.* and *Cylindrotheca sp.*, cyanobacteria, as well as autotrophic dinoflagellates (Courboulès *et al.* 2023). In addition, the pico- and nanophytoplankton abundances counted with flow cytometry also increased at this time of the experiment (Courboulès *et al.* 2023). Overall, a very good agreement was found between the response of the Chl-*a* concentration and phytoplankton abundances, measured by both microscopy and flow cytometry, during the entire experiment (Courboulès *et al.* 2023). The accumulation of phytoplankton biomass during the second part of the experiment in the runoff treatment was related to the strong increase in phytoplankton growth rate from d10, while the phytoplankton loss rate remained low until the end of the experiment. Consequently, the growth to loss ratio was significantly enhanced by more than ten times compared to that of the control. This suggests an uncoupling between phytoplankton growth and its loss factors, such as zooplankton and/or viruses, at that time in the experiment, possibly because phytoplankton grew too quickly compared to its predators. Nonetheless, this emphasises the potentially substantial structural impacts of terrestrial runoff on plankton communities and their intricate interactions within aquatic food webs, as recently documented in lakes (Strandberg *et al.* 2023).

The results of the present experiment suggest that the climate-change related intensification of terrestrial runoffs could temporarily alter metabolic and trophic indices of the water column of the lagoon during productive seasons (Trombetta *et al.* 2019), potentially shifting it towards heterotrophy and disrupting its trophic balance. Coupled with terrestrial runoff-induced shifts of microbenthic net community production towards heterotrophy (Liess *et al.* 2015), these alterations could interact with ongoing shifts occurring in the lagoon, such as the changes in trophic functioning towards mixotrophy and heterotrophy related to oligotrophication (Derolez *et al.* 2020b). Such consequences may also be seen in other Mediterranean lagoons, as turbidity and extreme flood events were reported to control phytoplankton abundance and phenology in oligotrophic Mediterranean coastal lagoons in Southern France and Corsica (Bec *et al.* 2011, Ligorini *et al.* 2022). Even though the results of the present study come from a single mesocosm experiment, implying that their generalisation should be implemented with care, they emphasise the importance of considering the effects of terrestrial runoffs on plankton-mediated processes in modelling projections of Mediterranean coastal waters under future climate scenarios.

**5 Data availability**

The data used in this paper are openly available in the SEANOE repository at https://www.seanoe.org/data/00861/97260/ (Soulié *et al.* 2023).

## 6 Acknowledgements

We thank David Parin, Romain Michel, Hervé Violette, Kilian Terrier, Inès Garcia, Valentin Kempf, and Paul Verzele, from MEDIMEER, for their help with the mesocosms and sensors setup, daily sampling, and analyses of chemical variables. We acknowledge Eftihis Nikiforakis for his help with discrete oxygen measurements and daily sampling. We also thank David Pecqueur, the SU/CNRS BioPIC Imaging and Cytometry platform and Barbara Marie, from the Observatoire Océanologique de Banyuls/Mer, for the cytometric and the dissolved organic carbon analyses, respectively. We are grateful to Valerio Caruso

from CNR-ISMAR for his valuable help in performing the total alkalinity analysis. We also thank the staff from the Puéchabon state forest for their help with practicalities during the sampling of soil. This work was part of the RESTORE project, funded by the French National Research Agency under the grant n°ANR-19-CE32-0013. C.C. was funded by the Transnational Access of the AQUACOSM-Plus project, which received funding from the European Union's Horizon 2020 research and innovation program under grant agreement n° 871081. As C.C. is a member of the JERICO-S3 project, which received funding from the

European Union's Horizon 2020 research and innovation program under grant agreements n° 871153 and n° 951799, her contribution to this work is also part of her contribution to JERICO-S3. A CC-BY public copyright license has been applied by the authors to the present document and will be applied to all subsequent versions up to the Author Accepted Manuscript arising from this submission, in accordance with the grant's open access conditions.

## 7 Author contribution

F.Vi. and B.M. designed the mesocosm experiment, and F.Vi., S.M., and B.M. managed it. F.Vi., J.C., M.H., S.M., F. Vo., C.C., F.J. and B.M. participated in the daily sampling of the experiment. C.C. performed the analysis of pH, $pCO_2$, and dissolved inorganic carbon, with the help of F.Vo. T.S. processed the sensor data, made all related analyses, and wrote the original draft of the manuscript, with inputs from all authors. All authors read and approved the final version of the manuscript.

## 8 Competing interests

The authors declare that they have no conflict of interest.

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
