# Peer review of "Simulated terrestrial runoff shifts the metabolic balance of a coastal Mediterranean plankton community toward heterotrophy"

_EGUsphere, 2023_

## Author Comment (AC1)

**Response to Referee Comment 1**

Dear Dr. Regaudie-de-Gioux,

We deeply thank you for your helpful comments, improving the quality and accuracy of the manuscript. Please find below the detailed explanations about how we have considered and answered each comment.

Concerning the modifications made on the revised version of the manuscript explained in the present document, please note that:
- additional sentences or words were highlighted in yellow
- deleted sentences or words were written

**1. A. Regaudie-de-Gioux**: The manuscript « Simulated terrestrial runoff shifts the metabolic balance of a coastal Mediterranean plankton community toward heterotrophy" evaluates the consequences of terrestrial runoff on planktonic communities in the coastal Mediterranean Thau Lagoon. For that the authors conducted in situ mesocosms experiment simulating terrestrial runoff by adding soil and measuring several chemical and biological parameters during almost 3 weeks.

The paper is easy to read, interesting and, I believe, deserve to be published in Biogeosciences.

M&Ms

L76: Experiments were performed during 18 days, why?

**The authors:** The duration of the experiment was set to 18 days to respond to multiple constraints. In one hand, the experiment needed to last long enough so that the responses of plankton at middle-term could be observed. We already performed multiple experiments in spring in Thau lagoon in the past (see Courboulès et al. 2022, Soulié et al. 2022a, 2022b) and reported that interesting dynamics as the occurrence of a phytoplankton bloom can occur up to almost 3 weeks after the beginning of the experiment, even in control mesocosms. In the other hand, the experiment from the present manuscript was conducted in May 2021, during severe lockdown restrictions due to the COVID-19 pandemics. Therefore, the experiment lasted 18 days, which was enough to monitor eventual responses, but not more due to methodological and external (lockdown) constraints.

Courboulès, J., Mostajir, B., Trombetta, T., Mas, S. & Vidussi, F. Warming disadvantages phytoplankton and benefits bacteria during a spring bloom in the Mediterranean Thau Lagoon. *Front. Mar. Sci.* **9**, 878938. https://doi.org/10.3389/fmars.2022.878938, 2022.

Soulié, T., F. Vidussi, J. Courboulès, S. Mas, and B. Mostajir. Metabolic responses of plankton to warming during different productive seasons in coastal Mediterranean waters revealed by in situ mesocosm experiments. *Sci. Rep.* **12**:9001. Doi: 10.1038/s41598-022-12744-x, 2022a.

Soulié, T., F. Vidussi, S. Mas, and B. Mostajir. Functional stability of a coastal Mediterranean plankton community during an experimental marine heatwave. *Front. Mar. Sci*. **9**:831496. Doi: 10.3389/fmars.2022.831496, 2022b.

**2. A. Regaudie-de-Gioux**: L87: Turn-over rate of 3.5d-1. Does that mean that every 3.5 days water from the surface goes down to the bottom of the mesocosm and goes then up to the surface?

**The authors:** The turn-over rate of 3.5 $d^{-1}$ of the pumps mixing the water column of the mesocosms means that every day, the entire water column of each mesocosms goes through the pump from the surface (i.e. a depth of 0.50 cm) to the bottom (i.e. a depth of 150 cm) of the mesocosms 3.5 times a day. In previous experiments, we used similar pumps and turnover rates to ensure homogenous water columns.

**3. A. Regaudie-de-Gioux**: L92: The data that you presented here are the average of 2 mesocosms for the control and 2 mesocosms for the treatment. It is quite critical to use these data with so few replicates…

**The authors:** The experiment was performed with 6 mesocosms in total, 3 controls and 3 mesocosms with terrestrial runoff addition. However, due to the malfunctioning of the pumps that were used to mix the water column of the mesocosms in 1 control mesocosm and 1 terrestrial runoff mesocosm, these 2 mesocosms are not real replicates as the mixing was different. An explanation in this regard is already given in Courboulès et al. (2023) but following your comment, a sentence was added in the Material and Method section (L92) to point out the low replication in this experiment.

L92: For each treatment, one mesocosm displayed considerable differences in biological, physical, and chemical parameters compared to the two other replicates of the same treatment, most probably because of the malfunctioning of the mixing pumps, and it was therefore removed from the analysis. Data are therefore presented as the mean of the two replicates for each treatment ± the range of observations. Thus, any interpretation of the presented data must take into account the low number of replication and be done cautiously.

Courboulès, J., Vidussi, F., Soulié, T., Nikiforakis, E., Heydon, M., Mas, S., Joux, F., and Mostajir, B. 2023: Effects of an experimental terrestrial runoff on the components of the plankton food web in a Mediterranean coastal lagoon. *Font. Mar. Sci*. **10**:1200757. https://doi.org/10.3389/fmars.2023.1200757.

**4. A. Regaudie-de-Gioux**: L101: "The mixture was left to mature for two weeks". Any refs? Why 2 weeks?

**The authors:** The maturation process was chosen to mimic natural residential time of the soil in the river water when natural terrestrial runoff processes occur in the Lagoon, such as during flash floods events (see Fouilland et al. 2012). The choice of using matured soil, and not fresh soil directly, was adapted from Müller

et al. (2018), and aimed at mimic natural processes during which most of the labile organic matter from the soil is degraded in the river water before reaching coastal waters. Then, different maturation times were tested during microcosm experiments about two months before the start of the mesocosm experiment, and a maturation step of 14 days was chosen as it represented the best compromise between mimicking natural runoff events and mimicking natural degradation processes of the organic matter.

Fouilland, E., Trottet, A., Bancon-Montigny, C., *et al*.: Impact of a river flash flood on microbial carbon and nitrogen production in a Mediterranean lagoon (Thau lagoon, France). *Est. Coast. Shelf Sci.* **113**:192-204. https://doi.org/10.1016/j.ecss.2012.08.004, 2012.

Müller, O., Seuthe, L., Bratbak, G., and Paulsen, M. L.: Bacterial response to permafrost derived organic matter input in an Arctic fjord. *Front. Mar. Sci.* **5**. https://doi.org/10.3389/fmars.2018.00263, 2018.

**5. A. Regaudie-de-Gioux**: L119: "Winkler Chl-a" , I think this is a typo error.

**The authors:** Thank you for noticing this error. We modify the sentence accordingly (L120).

L120: In addition, Chl-*a* fluorescence and oxygen sensor data were corrected using discrete high-performance liquid chromatography (HPLC) Chl-*a* and Winkler  DO measurements, respectively.

**6. A. Regaudie-de-Gioux**: L121: You said that you left at least during 6 h the DO bottle of fixation. This is quite a lot. Normally, we should leave the DO bottles, once fixed with R1 and R2, to settle the precipitate until reaching 1/3 of the bottle volume. If, for any logistical reason, we have to analyze latter the fixed DO bottles, It is advised to be kept in the dark and under water. Was that the case here?

**The authors:** Yes, samples were stored underwater and in the dark during the entire 6 hours of storage. Following your comment, this information was added to the text (L123). In addition, the protocol used for the Winkler titration is the one used routinely for dissolved oxygen measurement within the French Coastal Observation Service (Service d'Observation en Milieu Littoral, SOMLIT, https://www.somlit.fr/). The guidelines of SOMLIT regarding dissolved oxygen measurement with the Winkler method recommend between 6 hours and up to 1 month of storage after having added R1 and R2 (see the guidelines at https://www.somlit.fr/wp-content/uploads/2023/01/06-Protocole-national-O2-2023.pdf, p6). Additionally, Aminot & Kérouel (2004) report that fixed samples can be stored for at least 1 day. In all these regards, a storage duration of 6 hours was chosen and applied during our experiment.

L123: After at least 6 hr of fixation during which bottles were kept underwater and in the dark, the DO concentration in each bottle was measured with an automated Winkler titrator (Methrom 916-Ti-Touch) using a potentiometric titration method.

Aminot A., Kérouel R., 2004. Hydrologie des écosystèmes marins. Paramètres et analyses. Ed. Ifremer, 336p. ISBN: 2844331335, 9782844331335

**7. A. Regaudie-de-Gioux**: L135: Nutrient samples were stored at -20°C until analyses. However, it is not recommended to keep samples for silicate analyses at -20°C but rather at 4°C. Indeed, freezing can cause polymerization that can biased the analyses and specifically for samples from coastal, estuarine waters and with lower salinity (cf. Aminot and Kerouel). Considering that, I think the silicate data has to be analyzed with great precaution.

**The authors:** In line with your comment, we decided to remove the $SiO_2$ data from the manuscript, as the storage procedure of the samples cannot guarantee an unbiased measurement. Please note that, accordingly, some parts of the Materials and Methods (L132, L137) and Results (L246, L250, L320, L339) sections (notably parts of Figures 1 & 5) were removed.

L132: Each mesocosm was sampled daily using a 5 L Niskin water sampler at a depth of 1 m to monitor dissolved inorganic nutrients (nitrate + nitrite [$NO_2^-$+$NO_3^-$], ammonium [$NH_4^+$], and orthophosphate [$PO_4^{3-}$], and silicate [$SiO_2$]), dissolved organic carbon (DOC), particulate organic carbon (POC), and nitrogen (PON) concentrations; …

L137: Nitrate, nitrite, and orthophosphate, and silicate analyses were performed with an automated colorimeter

L246: Finally, the silicate concentrations ranged from 0.52 ± 0.15 µM to 0.79 ± 0.19 µM in the control treatment, they decreased on d2 before remaining relatively constant throughout the experiment (**Fig. 1m**). They were significantly higher in the terrestrial runoff treatment during the entire experiment by 214% (**Table 1**).

[Figure]

**Figure 1. Daily average temperature (a), salinity (b), daily light integral (DLI, c), pH (d), pCO₂ (e), dissolved organic carbon concentrations (DOC, f), dissolved inorganic carbon concentrations (DIC, g), particulate organic carbon concentrations (POC, h), particulate organic nitrogen concentrations (PON, i), nitrate + nitrite concentrations (NO₂⁻+NO₃⁻, j), ammonium concentrations (NH₄⁺, k), ==and== orthophosphate concentrations (PO₄³⁻, l) in the control (blue) and terrestrial runoff (gold) treatments. Error bars represent the range of the observations.**

**Table 1. Summary table of the statistical comparison and the % relative change between the terrestrial runoff and the control treatments. The significance level was set to 0.05 and significant P-values, as well as their corresponding relative change, were highlighted in bold. When a RM-ANOVA was performed, its F value was given in brackets, and when a Kruskal-Wallis was performed instead, "KW" was indicated.**

| Parameter | Period | P-value | % difference |
|---|---|---|---|
| Temperature | 2-18 | 0.64 (KW) | -0.2 |
| Salinity | 2-18 | **$< 1\times10^{-4}$** ($F_{1,16}=1035$) | **-0.7** |
|  | 2-18 | **$1.4\times10^{-3}$** (KW) | **-43.3** |

| | | | |
|---|---|---|---|
| DLI | 2-11 | **9.1×10⁻⁴** (KW) | **-51.6** |
| | 12-18 | **1.4×10⁻³** ($F_{1,6}=32.9$) | **-27.3** |
| pH | 2-18 | **3×10⁻⁴** ($F_{1,9}=32.4$) | **-0.4** |
| pCO$_2$ | 2-18 | **4×10⁻⁴** ($F_{1,9}=30.5$) | **8.9** |
| DOC | 2-18 | 0.27 (KW) | 0.4 |
| DIC | 2-18 | **7×10⁻⁴** ($F_{1,9}=25.3$) | **1.3** |
| | 2-18 | **2.2×10⁻⁶** ($F_{1,16}=11.5$) | **27.8** |
| POC | 2-12 | **1.1×10⁻⁸** ($F_{1,10}=27.9$) | **49.3** |
| | 12-18 | 0.621 (KW) | -2.0 |
| | 2-18 | **0.001** (KW) | **18.8** |
| PON | 2-12 | **1.6×10⁻⁵** ($F_{1,10}=12.9$) | **32.3** |
| | 12-18 | 0.474 ($F_{1,6}=0.7$) | -2.7 |
| NO$_2^-$ + NO$_3^-$ | 2-18 | 0.75 ($F_{1,16}=0.1$) | 1.3 |
| NH$_4^+$ | 2-18 | **3.2×10⁻⁴** (KW) | **486.5** |
| PO$_4^{3-}$ | 2-18 | **0.02** ($F_{1,16}=6.8$) | **18.0** |
| | 10-13 | **8.4×10⁻³** ($F_{1,3}=38.7$) | **62.7** |
|  |  |  |  |
| GPP | 2-17 | 0.37 (KW) | 16.1 |
| | 9-14 | **1.1×10⁻³** ($F_{1,5}=44.7$) | **36.6** |
| | 12-17 | 0.08 ($F_{1,15}=4.8$) | 24.5 |
| GPP : Chl-$a$ | 2-17 | **0.02** (KW) | **312.1** |
| R | 2-17 | **<1×10⁻⁴** ($F_{1,15}=38.4$) | **45.7** |
| | 2-11 | **2×10⁻⁴** ($F_{1,10}=32.7$) | **52.5** |
| GPP : R | 2-17 | **7×10⁻⁴** ($F_{1,15}=18.4$) | **-17.6** |
| | 2-10 | **<1×10⁻⁴** ($F_{1,15}=82.5$) | **-32** |
| F$_v$ : F$_m$ | 2-17 | 0.94 ($F_{1,17}=0.01$) | 0.4 |
| | 8-11 | **3.7×10⁻³** ($F_{1,5}=68.7$) | **43.0** |
| R : Chl-$a$ | 2-17 | **7.5×10⁻⁴** (KW) | **419.9** |
| R : Bacteria | 2-17 | 0.16 (KW) | 26.9 |
| | 9-14 | **2×10⁻⁴** ($F_{1,5}=22.1$) | **154.2** |
| | 2-18 | 0.183 (KW) | 15.0 |
| Bacterial abundances | 2-8 | **1.3×10⁻⁴** ($F_{1,6}=37.2$) | **59.0** |
| | 9-14 | **2.5×10⁻³** ($F_{1,5}=24.2$) | **-47.0** |
| | 15-18 | **6.7×10⁻³** ($F_{1,3}=22.7$) | **51.0** |
| | 2-18 | **1.2×10⁻³** ($F_{1,17}=14.9$) | **-30.2** |
| Chl-$a$ | 2-11 | **1.6×10⁻⁴** (KW) | **-70.2** |
| | 12-18 | 0.89 ($F_{1,8}=0.02$) | 4.2 |
| | 2-17 | 0.86 ($F_{1,15}=0.03$) | 110.6 |
| Growth rate (μ) | 2-11 | **0.02** ($F_{1,8}=7.5$) | **-52.8** |
| | 12-17 | **3.0×10⁻⁴** ($F_{1,5}=77.9$) | **298.7** |
| Loss rate (L) | 2-17 | **4.7×10⁻³** ($F_{1,15}=11$) | **-32.1** |
| | 3-14 | **6×10⁻⁴** ($F_{1,11}=22.3$) | **-60.0** |
| μ : L ratio | 2-17 | **0.02** ($F_{1,15}=7.3$) | **305.4** |
| | 11-18 | **1×10⁻⁴** ($F_{1,5}=115$) | **550.3** |

L339:

[Figure]

**Figure 5. Correlation matrix based on Spearman's correlations between the log response ratio (LRR) of phytoplankton processes, community metabolism, and environmental variables. Only significant (*p* < 0.05) correlations are shown in the matrix. Green illustrates positive correlations and purple negative correlations. (GPP: Gross Primary Production, R: Respiration, Chl-*a*: Chlorophyll-*a*, μ: Growth rate, L: Loss rate, DLI: Daily Light Integral, DOC: Dissolved Organic Carbon, Dissolved Inorganic Carbon, POC + PON: Particulate Organic Carbon + Nitrogen).**

**8. A. Regaudie-de-Gioux**: How do you think, by the end of the experiment, the diminution of mesocosms volume can affect planktonic communities and thus your analyses?

**The authors:** Initially, each mesocosm was filled with 2200 L of lagoon water. Then, we sampled a total of 510 L for each mesocosm, representing 23% of the initial volume. As shown in Spivak et al. (2011), mesocosm volume generally does not play a huge role in the response of plankton (at least phytoplankton). Hence, we believe that the diminution of mesocosm volume by less than a quarter by the end of the

experiment did not affect our analyses, taking also into account that the diminution of the water volume is gradual.

Spivak, A. C., Vanni M. J., and Mette E. M.: Moving on up: can results from simple aquatic mesocosm experiments be applied across broad spatial scales? *Fresh. Biol.* **56**: 279-291, https://doi.org/10.1111/j.1365-2427.2010.02495.x, 2011

**9. A. Regaudie-de-Gioux**: L175-185: What is the % of error of this method? Its reproducibility and inter-repeatability?

**The authors:** This method (Soulié et al., 2021) is based on the diel oxygen technique (Staehr et al., 2010), and enables to estimate GPP, R, and NCP from high-frequency measurements of dissolved oxygen concentration. As this method employs sensor measurements and mathematical calculations to estimate metabolic parameters, its errors, reproducibility, and inter-repeatability are related to those of the sensors used to acquire data (i.e., the method applied multiple times to the same data will always yield exactly the same results). Therefore, the main uncertainties come from: the accuracy of sensor measurements of dissolved oxygen, the estimation of the air-water exchange coefficient, and the sampling frequency applied to the sensors.

- Accuracy of sensor measurements:

The method relies on accurate measurements of dissolved oxygen concentrations, which were performed with oxygen optodes. The principles of such sensors are detailed in Bittig et al. (2018), as well as uncertainties and errors embedded with the sensors. The main caveat is the time-dependent response drift. To correct for this potential bias, optodes were calibrated before and after deployment, and sensor data were corrected with discrete Winkler measurements.

Bittig, H. C., Körtzinger A., Neill C., van Ooijen E., Plant N. J., Hahn J., Johnson K. S., Yang B., and Emerson S. R.: Oxygen optode sensors: Principle, characterization, calibration, and application in the ocean. *Front. Mar. Sci.* **4**:429, https//doi.org/10.3389/fmars.2017.00429, 2018

- Estimation of the air-water exchange coefficient:

The pace at which oxygen exchanges with the atmosphere over time can be estimated as directly proportional to the oxygen deficit, exemplified by the variance between dissolved oxygen (DO) and the saturation level of oxygen. To gauge the transfer of oxygen from the atmosphere to the water surface, the piston velocity coefficient denoted as k, functioning as a proportional constant, is intricately linked to factors such as surface turbulence, internal mixing, water viscosity, and temperature. In practical applications, the determination of k involves the consideration of wind speed at a height of 10 meters above the water surface. However, this calculation proves impractical in enclosed mesocosms covered by a dome, as they are not directly exposed to turbulence induced by wind. Despite the absence of direct wind effects, attributing k to zero in mesocosms is not possible due to the movement of water mass within the mesocosms, influenced by external waves surrounding it and mixing induced by the pump. Hence, the estimation of the air-water exchange coefficient is the most important uncertainty of the method. Analyses of sensitivity of the method to the estimation of k was done by Staehr et al. (2010) and Soulié et al. (2021), both of which revealed that GPP seemed unaffected by uncertainties in K while R and NCP were significantly changed depending on k. In the present manuscript, k was chosen from the literature (Alcaraz et al., 2001) and

corrected for temperature and salinity. We are fully aware of the uncertainty associated with the air-water exchange coefficient in our metabolism estimates, but want to stress out the fact that, as the same k was applied to both control and runoff mesocosms, this uncertainty did not affect the differences in metabolism between treatments, thus preventing from altering the reported effects of the runoff on the metabolic parameters.

Staehr, P. A., Bade D., Van de Bogert M. C., and others.: Lake metabolism and the diel oxygen technique: State of the science. *Limnol. Oceanogr. Methods* **8**: 628-644, https://doi.org/10.4319/lom.2010.8.628, 2010

Soulié, T., Mas, S., Parin, D., Vidussi, F., and Mostajir, B. : A new method to estimate planktonic oxygen metabolism using high-frequency sensor measurements in mesocosm experiments and considering daytime and nighttime respirations. *Limnol. Oceanogr. Methods* **19**:303-316. https://doi.org/10.1002/lom3.10424, 2021a

- Sampling frequency applied to the sensors:

Another uncertainty related to the method is the choice of an adequate sampling frequency. Indeed, valuable insights from dissolved oxygen (DO) data may be overlooked when the sampling frequency is too low, resulting in disparities in metabolic assessments. Conversely, a sampling frequency that is too high can generate an extensive dataset that is not always unnecessary to yield additional information. Previous research suggests that a sampling frequency of 30 minutes is sufficient for obtaining reliable daily metabolic estimates during field observations of lakes (Staehr et al. 2010). The same statistical power analysis was done in the case of mesocosm experiment (Soulié et al., 2021a), and it appeared that a sampling frequency of one measurements every 1 minutes, as done in the present experiment, is largely sufficient to obtain powerful estimates for a deployment of 18 days.

**10. A. Regaudie-de-Gioux**: Figure 1: How do you explain the great drop of DLI at day 12 and 13?

**The authors:** The important decrease of available light at days 12 and 13 were due to bad weather conditions, as weather conditions on May 15[th] (d12) and May 16[th] (d13), 2021, were extremely cloudy (https://www.historique-meteo.net/france/languedoc-roussillon/sete/2021/05/15/, https://www.historique-meteo.net/france/languedoc-roussillon/sete/2021/05/16/) .

**11. A. Regaudie-de-Gioux**: Figure 2: BA had opposite trends between control and treatment. How do you explain it? Why did you not analyze the phytoplanktonic abundance?

**The authors:** First, regarding phytoplankton abundances, phytoplankton community was analyzed through microscopy and flow cytometry. These results are already presented in Courboulès et al. (2023) which was already cited in the present manuscript (L69). As most phytoplankton groups displayed a dynamic very similar to the Chlorophyll-*a* concentrations dynamics, and as the focus of the present manuscript was more on the functional than the compositional responses of the plankton community and that these data are already published, we decided not to present them.

Regarding the bacterial abundance data, we agree that it displayed interesting dynamics in both the control and the terrestrial runoff treatment. Bacteria abundance was favored by the treatment during the first 8 days of the experiment. This is congruent with other studies investigating the effects of terrestrial matter addition during mesocosm experiments (Guadayol et al., 2009; Rasconi et al., 2015; Liess et al., 2016), and even after natural flood events in Thau Lagoon (Fouilland et al., 2011). For the present experiment, Courboulès et al. (2023) proposed that bacteria initially took advantage of the input of particulate matter through the runoff simulation. During the second part of the experiment (days 9 to 14), bacteria abundance displayed a negative response in the terrestrial runoff compared to the control. Bacteria could have been disadvantaged by the fact that most of the particulate matter inputs in the terrestrial runoff treatment had solubilized by then, and/or by the fact that phytoplankton biomass was really low at this time of the experiment, resulting in lower dissolved organic carbon availability from phytoplankton to bacteria. Additionally, Courboulès et al. (2023) proposed higher grazing of bivalvia larvae on bacteria to explain this negative effect on bacteria. Finally, at the end of the experiment, bacteria abundance started to increase again, with a positive effect of the runoff treatment from day 15 to 18. This increase occurred concomitantly with a huge increase in phytoplankton biomass, suggesting that higher phytoplankton biomass may have fueled bacteria with dissolved organic carbon.

**12. A. Regaudie-de-Gioux**: Figure4: Did you check CR/Chl or CR/BA?

**The authors:** Thank you for your suggestion. We agree that these results can be useful and thus we added both CR/Chl-*a* and CR/BA in the Results section (L306, L320). As a consequence, the figure 4 was modified accordingly, as well as Table 1 (L320) in which we added the results of the statistical tests we performed with CR/Chl-*a* and CR/BA. Some parts in the Discussion section were also modified accordingly (L400) as follows:

L306: When the R was normalised by the daily Chl-*a* concentration, it was significantly higher in the terrestrial runoff treatment than in the control by an average of 420% throughout the experiment (**Fig. 4f**).

Finally, when R was normalised by total bacterial abundance (**Fig. 4g**), it was not significantly different between treatments apart from d9 to d14 during which it was significantly higher by an average of 154% in the terrestrial runoff treatment than in the control (**Table 1**).

[Figure]

**Figure 4. Daily average gross primary production (GPP, a), GPP normalised by chlorophyll-*a* (GPP:Chl-*a*, b), community respiration (R, c), GPP : R ratio (d),  maximum quantum yield (F$_v$ : F$_m$) of photosystem II (PSII) (e), R normalised by chlorophyll-a (R:Chl-a, f), and R normalised by bacterial abundance (R:Bacteria, g) in the control (blue) and terrestrial runoff (gold) treatments. Error bars represent the range of the observations. Note that GPP and R could not be estimated on d18 owing to the lack of a complete oxygen cycle.**

L320:

**Table 1. Summary table of the statistical comparison and the % relative change between the terrestrial runoff and the control treatments. The significance level was set to 0.05 and significant P-values, as well as their corresponding relative change, were highlighted in bold. When a RM-ANOVA was performed, its F value was given in brackets, and when a Kruskal-Wallis was performed instead, "KW" was indicated.**

| Parameter | Period | P-value | % difference |
|---|---|---|---|
| Temperature | 2-18 | 0.64 (KW) | -0.2 |
| Salinity | 2-18 | **$< 1 \times 10^{-4}$ ($F_{1,16}$=1035)** | **-0.7** |
| | 2-18 | **$1.4 \times 10^{-3}$ (KW)** | **-43.3** |

| | | | |
|---|---|---|---|
| DLI | 2-11 | **$9.1\times10^{-4}$ (KW)** | **-51.6** |
| | 12-18 | **$1.4\times10^{-3}$ ($F_{1,6}=32.9$)** | **-27.3** |
| pH | 2-18 | **$3\times10^{-4}$ ($F_{1,9}=32.4$)** | **-0.4** |
| pCO$_2$ | 2-18 | **$4\times10^{-4}$ ($F_{1,9}=30.5$)** | **8.9** |
| DOC | 2-18 | 0.27 (KW) | 0.4 |
| DIC | 2-18 | **$7\times10^{-4}$ ($F_{1,9}=25.3$)** | **1.3** |
| POC | 2-18 | **$2.2\times10^{-6}$ ($F_{1,16}=11.5$)** | **27.8** |
| | 2-12 | **$1.1\times10^{-8}$ ($F_{1,10}=27.9$)** | **49.3** |
| | 12-18 | 0.621 (KW) | -2.0 |
| PON | 2-18 | **0.001 (KW)** | **18.8** |
| | 2-12 | **$1.6\times10^{-5}$ ($F_{1,10}=12.9$)** | **32.3** |
| | 12-18 | 0.474 ($F_{1,6}=0.7$) | -2.7 |
| NO$_2^-$ + NO$_3^-$ | 2-18 | 0.75 ($F_{1,16}=0.1$) | 1.3 |
| NH$_4^+$ | 2-18 | **$3.2\times10^{-4}$ (KW)** | **486.5** |
| PO$_4^{3-}$ | 2-18 | **0.02 ($F_{1,16}=6.8$)** | **18.0** |
| | 10-13 | **$8.4\times10^{-3}$ ($F_{1,3}=38.7$)** | **62.7** |
|  |  |  |  |
| GPP | 2-17 | 0.37 (KW) | 16.1 |
| | 9-14 | **$1.1\times10^{-3}$ ($F_{1,5}=44.7$)** | **36.6** |
| | 12-17 | 0.08 ($F_{1,15}=4.8$) | 24.5 |
| GPP : Chl-*a* | 2-17 | **0.02 (KW)** | **312.1** |
| R | 2-17 | **$<1\times10^{-4}$ ($F_{1,15}=38.4$)** | **45.7** |
| | 2-11 | **$2\times10^{-4}$ ($F_{1,10}=32.7$)** | **52.5** |
| GPP : R | 2-17 | **$7\times10^{-4}$ ($F_{1,15}=18.4$)** | **-17.6** |
| | 2-10 | **$<1\times10^{-4}$ ($F_{1,15}=82.5$)** | **-32** |
| F$_v$ : F$_m$ | 2-17 | 0.94 ($F_{1,17}=0.01$) | 0.4 |
| | 8-11 | **$3.7\times10^{-3}$ ($F_{1,5}=68.7$)** | **43.0** |
| ==R : Chl-*a*== | ==2-17== | ==**$7.5\times10^{-4}$ (KW)**== | ==**419.9**== |
| ==R : Bacteria== | ==2-17== | ==0.16 (KW)== | ==26.9== |
| | ==9-14== | ==**$2\times10^{-4}$ ($F_{1,5}=22.1$)**== | ==**154.2**== |
| Bacterial abundances | 2-18 | 0.183 (KW) | 15.0 |
| | 2-8 | **$1.3\times10^{-4}$ ($F_{1,6}=37.2$)** | **59.0** |
| | 9-14 | **$2.5\times10^{-3}$ ($F_{1,5}=24.2$)** | **-47.0** |
| | 15-18 | **$6.7\times10^{-3}$ ($F_{1,3}=22.7$)** | **51.0** |
| Chl-*a* | 2-18 | **$1.2\times10^{-3}$ ($F_{1,17}=14.9$)** | **-30.2** |
| | 2-11 | **$1.6\times10^{-4}$ (KW)** | **-70.2** |
| | 12-18 | 0.89 ($F_{1,8}=0.02$) | 4.2 |
| Growth rate (μ) | 2-17 | 0.86 ($F_{1,15}=0.03$) | 110.6 |
| | 2-11 | **0.02 ($F_{1,8}=7.5$)** | **-52.8** |
| | 12-17 | **$3.0\times10^{-4}$ ($F_{1,5}=77.9$)** | **298.7** |
| Loss rate (L) | 2-17 | **$4.7\times10^{-3}$ ($F_{1,15}=11$)** | **-32.1** |
| | 3-14 | **$6\times10^{-4}$ ($F_{1,11}=22.3$)** | **-60.0** |
| μ : L ratio | 2-17 | **0.02 ($F_{1,15}=7.3$)** | **305.4** |
| | 11-18 | **$1\times10^{-4}$ ($F_{1,5}=115$)** | **550.3** |

L400: In the present study, bacterial abundance was significantly enhanced by the runoff during the first part of the experiment (d2-d9), which is congruent with the higher respiration at that time. ==This suggests==

that higher bacterial abundances are certainly responsible for the higher R reported in the runoff treatment during the first part of the experiment. However, bacterial abundances then significantly decreased during the middle of the experiment (d9-d14) in the runoff treatment, while respiration remained significantly higher than in the control treatment, resulting in a positive response of R normalised by bacterial abundance at this time of the experiment. This suggests  that respiration was mostly not sustained by bacteria at that time of the experiment, but by other biological compartments instead. Because Chl-*a* was still strongly depressed by the runoff during this period of the experiment, resulting in extremely high R:Chl-*a* rates, the hypothesis of an increase in phytoplankton respiration is not plausible.

**13. A. Regaudie-de-Gioux**: L382-387: Considering the mixing of the mesocosms waters (turn-over of 3.5 days), do you think that sedimentation is important?

**The authors:** Pumps were installed inside the mesocosms to prevent important stratification, but not to prevent sedimentation. We know that some sedimentation occurred because the mesocosms have sediment traps and sedimented material was collected at multiple times during this experiment. These sediment trap samples are not fully analyzed yet, and it is difficult to state now which part of the community/particles sedimented, and at what speed, however it is clear that the quantity of sedimented material was much higher in the runoff treatment than in the control. In agreement with your comment, a sentence was added in the Discussion to state that the role of sedimentation in the responses we observed is at this time speculative (L391).

L391: An alternative hypothesis is that the high quantity of particulate matter added through the simulated runoff induced a strong sedimentation of a part of the phytoplankton community toward the bottom of the mesocosm enclosures (Kiorboe *et al.* 1990). This sedimentation could have partly contributed to the mismatch between GPP and Chl-*a*, as sedimented phytoplankton could have continued to produce oxygen, while being undetected by both manual and sensor monitoring of Chl-*a*. Such sedimentation has already been suggested after heavy loadings of terrestrial matter during a natural flash flood event in Thau Lagoon, during which most of the microbial production may have been exported through sedimentation (Fouilland *et al.* 2012). Nonetheless, it should be noted that the samples of sedimented material in the sediment traps are not fully analyzed yet, thus preventing to characterize the role of sedimentation in the responses of GPP and Chl-*a* with certainty.

**14. A. Regaudie-de-Gioux**: L390: There is no relationship between CR and BA, so we cannot think that planktonic bacterial respiration represent a major part of community respiration here.

**The authors:** Indeed, no significant correlation was found between bacterial abundances and R (cf Figure 5). However, this correlation was performed with data from the entire experiment. During the first part of the experiment (days 2-8), both bacterial abundances and R increased significantly in the terrestrial runoff treatment compared to the control. This suggests that bacteria indeed played an important part in community R at this time of the experiment (from days 2 to 8). Nonetheless, we agree that this is not the case after day 8 (from days 9 to 14), as bacterial abundances decreased but R stays high. We added precision in the Discussion section regarding this (L400).

L400: In the present study, bacterial abundance was significantly enhanced by the runoff during the first part of the experiment (d2-d9), which is congruent with the higher respiration at that time. This suggests that higher bacterial abundances are certainly responsible for the higher R reported in the runoff treatment during the first part of the experiment. However, bacterial abundances then significantly decreased during the middle of the experiment (d9-d14) in the runoff treatment, while respiration remained significantly higher than in the control treatment, resulting in a positive response of R normalised by bacterial abundance at this time of the experiment. This suggests  that respiration was mostly not sustained by bacteria at that time of the experiment, but by other biological compartments instead. Because Chl-*a* was still strongly depressed by the runoff during this period of the experiment, resulting in extremely high R:Chl-*a* rates, the hypothesis of an increase in phytoplankton respiration is not plausible.

**15. A. Regaudie-de-Gioux**: L407: Do you think that Mediterranean coastal lagoons are really net $O_2$ producers today? Any refs?

**The authors:** The net oxygen balance of coastal Mediterranean lagoons is highly variable, and depend notably on the location of the lagoon, the season, the trophic status etc. Some research highlighted that some Mediterranean lagoons acted as net $O_2$ producers, depending notably on the year (e.g., Bas-Sylvestre et al. 2020). But we believe that this could change due to the effects of climate change (e.g., warming, runoffs…) which could turn them net heterotrophic more often. Regarding Thau Lagoon in particular, the net oxygen balance appears to mainly be driven by seasonality ('Malaigues' anoxic crisis) and human activities (oyster farming). Therefore, we think that not all Mediterranean lagoons act as net oxygen producers. We modified the sentence accordingly (L419).

L419: Therefore, the present experiment shows, for the first time to our knowledge in Mediterranean coastal lagoons, that terrestrial runoffs could potentially shift coastal Mediterranean lagoons, such as Thau Lagoon, from being net oxygen producer in spring to net oxygen sink.

Bas-Sylvestre, M., Qunitana, X. D., Compte J., Gascon S., Boix D., Anton-Pardo M., and Obrador B.: Ecosystem metabolism dynamics and environmental drivers in Mediterranean confined coastal lagoons. *Est. Coast Shelf Sci*. **245**: 106989, https://doi.org/10.1016/j.ecss.2020.106989, 2020.

**16. A. Regaudie-de-Gioux**: Supplement information

Rnight = (mean of NCP during night period)*duration of night period*60 (Soulié et al. 2021.

**The authors:** Thank you for noticing this difference. In Soulié et al. (2021), the equation for Rnight is given with duration of the "Night period", and it is said that "the Night period refers to the Negative NCP period". We modified the equation so that it appears the same in the Supplementary Information as in Soulié et al. (2021).

---

## Author Comment (AC2)

**Response to Referee Comment 2**

Dear Dr. Cibic,

We deeply thank you for your helpful comments, improving the quality and accuracy of the manuscript. Please find below the detailed explanations about how we have considered and answered each comment.

Concerning the modifications made on the revised version of the manuscript explained in the present document, please note that:
- additional sentences or words were highlighted in blue
- deleted sentences or words were written

Please note that modifications made in response to the comments of the other referee (Referee 1, A. Regaudie-de-Gioux) are also indicated on the document, with the added parts highlighted in yellow and deleted parts .

**1. T. Cibic**: The work by Soulié et al. "Simulated terrestrial runoff shifts the metabolic balance of a coastal Mediterranean plankton community toward heterotrophy" investigated the consequences of terrestrial runoff on plankton communities and some biological processes in the Thau lagoon on the Mediterranean coast. The results come from an in situ mesocosm experiment in which terrestrial runoff was simulated by adding soil and various chemical and biological variables were analyzed for 18 days.

The paper is well written, focused and interesting and should be published in Biogeosciences after revision.

A major criticism of this work is the fact that no real phytoplankton data are presented. All results were derived from Chl-a sensor data, including phytoplankton growth and loss rates. I know this is becoming more common and accepted lately, but similarly to Chl-*a* fluorescence and oxygen sensors data, for which some calibrations were done, actual phytoplankton counts and identifications should have been done on at least some samples to check if there is a match between Chl-*a* and microphytoplankton. Since flow cytometry was used in the paper to estimate heterotrophic bacterial abundance, the same method could have been used to assess the smaller phototrophic picoplankton. It is a pity that there is no information on the actual composition of the phytoplankton, as this is the topic of the article. If the authors have these (already published or unpublished) results, I think it would be a great addition to this publication to at least mention them in the discussion.

**The authors:** We agree that information on the composition of the phytoplankton community is important to complement chlorophyll-*a* data. In our experiment, phytoplankton community was analyzed by microscopy (identification and counting) and by flow cytometry. All these data are presented and published in Courboulès et al. (2023), which is cited at multiple times in the manuscript. These data show a very good agreement with chlorophyll-*a*, especially microphytoplankton, diatoms, identified and counted by microscopy, and nanophytoplankton, counted by flow cytometry, and this good agreement was already reported in Courboulès et al. (2023). As these data are already published and well-discussed in Courboulès et al. (2023), we feel that it is not necessary to present them in details again in the present manuscript. However, we agree that it should be mentioned in the Discussion section. Hence, we added this information in the Discussion section (L363, L377).

L363: "The phytoplankton community investigated in the present study was typical of Thau Lagoon in spring (Trombetta *et al.* 2019), mainly composed of diatoms, cryptophytes, and small nano- and picophytoplankton (Courboulès *et al.* 2023). The  negative effect of light limitation induced by the runoff on phytoplankton biomass is consistent with a mesocosm experiment performed in the Baltic Sea…"

L377: "As mentioned earlier, Chl-*a* strongly increased during the second part of the experiment in the runoff treatment. This positive response was mainly due to an increase in the abundance of diatoms, mainly *Chaetoceros sp.* and *Cylindrotheca sp.*, cyanobacteria, and autotrophic dinoflagellates (Courboulès *et al.* 2023). In addition, the pico- and nanophytoplankton abundances counted with flow cytometry also increased at this time of the experiment (Courboulès *et al.* 2023). Overall, a very good agreement was found between the response of the Chl-*a* concentration and phytoplankton abundances, measured by both microscopy and flow cytometry, during the entire experiment (Courboulès *et al.* 2023).  The accumulation of phytoplankton biomass during the second part of the experiment in the runoff treatment was related to the strong increase in phytoplankton growth rate from d10"

**2. T. Cibic: Introduction**

I think it would be easier for the reader if some scientific questions or hypotheses were added at the end of the introduction section to structure the discussion.

**The authors:** We agree with this comment. Therefore, a sentence was added presenting some hypotheses at the end of the Introduction section (L77).

L77: "In the present study, high-frequency data from automated sensors immersed in the mesocosms were used to estimate GPP, R, μ and L in every mesocosm, and assess how both the metabolic and trophic indices of the community responded to the simulated runoff. Manual sampling was performed to assess dissolved and particulate materials as well as photosynthetic efficiency and carbonate system parameters. We hypothesized that (1) the metabolic index would be shifted by the runoff toward heterotrophy through light reduction and inputs of organic matter, and that (2) the terrestrial runoff would affect the trophic index by creating imbalance between phytoplankton and its factors of loss."

**3. T. Cibic: M&Ms**

Experimental design

It is not stated what the final volume in each mesocosm is. It is also not clear if the mesocosms are sealed at the top and what the bottom is like. Are the mesocosms open or sealed at the bottom? Are they floating? Please add this information.

**The authors:** We agree that this information was not clear, so we added some parts in the M&Ms section (L89, L100). We explained that, initially, each mesocosm was filled with 2200 L of natural lagoon water. Then, a total of 510 L for each mesocosm was sampling during the experiment, representing 23% of the initial volume. Mesocosms were sealed at the bottom by a sediment trap, floating, and attached on a floating pontoon, such as represented in the figure below that we added in Supplementary Information.

[Figure]

L89: "Six mesocosms were established in the lagoon. Each mesocosm consisted of a bag, sealed at the bottom, made of nylon-reinforced 200 μm thick vinyl acetate polyethylene film which was 280 cm high and 120 cm wide (Insinööritoimisto Haikonen Ky, Sipoo, Finland). Each mesocosm was equipped with a sediment trap at the bottom. A schematic representation of the mesocosm set-up can be found in Soulié *et al.* (2021) and in **Supplementary Information**."

L100:"Throughout the experiment, a total of 510 L was sampled for each mesocosm, representing 23% of the initial volume of the mesocosms."

**4. T. Cibic**: L76: The experiments were conducted for 18 days. In my opinion, this is a very long time to test the effects of a flash flood on the coastal area. There is a strong mixing effect by the seawater, and even in a confined area like a lagoon, the terrestrial runoff will eventually be diluted in a few days. The choice of the duration of the experiment should be discussed.

**The authors:** We agree with the fact that flash floods do not last as long as 18 days. However, the goal of the experiment was not only to study the short-term impact of flash floods and terrestrial runoffs, but also the longer-lasting consequences of such events, even when terrestrial matter has already sedimented and/or

dissolved. As explained to the Referee 1, the duration of the experiment was set to 18 days to respond to multiple objectives. As explained above, the experiment needed to last long enough so that the responses of plankton at middle-term could be monitored. Indeed, we have already performed multiple mesocosm experiments in spring in Thau lagoon in the past (see Courboulès et al. 2022, Soulié et al. 2022a, 2022b) and reported that interesting dynamics as the occurrence of a phytoplankton bloom can occur up to almost 3 weeks after the beginning of the experiment, even in control mesocosms. Some sentences were added in the Material and Method section regarding the choice of the duration of the experiment (L84).

L84: "The duration of the experiment was set as 18 days to be able to monitor the responses of plankton at medium-term (multiple days to weeks), as interesting dynamics were already reported in control treatments during other experiments in Thau Lagoon up to almost 3 weeks after the start of the experiment (Courboulès et al. 2021, Soulié et al. 2022a), while coping with COVID-19 pandemics restrictions."

Courboulès, J., Mostajir, B., Trombetta, T., Mas, S. & Vidussi, F. Warming disadvantages phytoplankton and benefits bacteria during a spring bloom in the Mediterranean Thau Lagoon. *Front. Mar. Sci.* **9**, 878938. https://doi.org/10.3389/fmars.2022.878938, 2022.

Soulié, T., F. Vidussi, J. Courboulès, S. Mas, and B. Mostajir. Metabolic responses of plankton to warming during different productive seasons in coastal Mediterranean waters revealed by in situ mesocosm experiments. *Sci. Rep.* **12**:9001. Doi: 10.1038/s41598-022-12744-x, 2022a.

Soulié, T., F. Vidussi, S. Mas, and B. Mostajir. Functional stability of a coastal Mediterranean plankton community during an experimental marine heatwave. *Front. Mar. Sci.* **9**:831496. Doi: 10.3389/fmars.2022.831496, 2022b.

**5. T. Cibic:** L104: Why was the sampled soil left to mature for two weeks? Why was this step necessary? In a flash flood there is no such step, the soil is washed away directly by the heavy rain and transported into a river and eventually into an estuary. Flash floods do not last 14 days, this is unclear and should be better explained.

**The authors:** The maturation process, during which the soil of the Puéchabon forest was mixed with water from the Vène river, aimed at recreating the natural transport of soil that can be transported for several weeks by rivers before reaching coastal waters. The goal of this was to mimic naturally occurring processes, as during natural terrestrial runoff, the most labile compounds of the soil can be degraded or transformed during their transportation by river water (Lobbes et al. 2000, Müller et al. 2018). We agree that the choice of this maturation step should be discussed. Therefore, we added some sentence in the Discussion section regarding this (L375).

L375: "Finally, Deininger *et al.* (2016) used a resin in their soil extraction procedure, yielding higher inorganic and organic nutrient concentrations in their extract compared to the protocol performed in the present study but being farther from natural terrestrial runoffs (Scharnweber *et al.* 2021). In the present experiment, a maturation step of the soil in river water of 14 days was performed, aiming at mimicking processes naturally occurring during the transportation of soil to coastal waters during terrestrial runoffs, such as the degradation of the most labile organic compounds (Müller *et al.* 2018). This duration of maturation can be considered as a long residence time in river water, regarding the fact that flash floods in the Mediterranean region are usually faster. Therefore, it can be supposed that the terrestrial matter added in the present study contained lower levels of labile organic compounds than what can be found during

flash floods. This emphasises the need for extreme caution when comparing experimental studies investigating terrestrial runoff effects because protocols are often different from one study to another."

Lobbes, J. M., Fitznar, H. P., and Kattner, G.: Biogeochemical characteristics of dissolved and particulate organic matter in Russian rivers entering the Arctic Ocean. *Geochemica et Cosmochimica Acta* **64**(17):2973-2983. https://doi.org/10.1016/S0016-7037500°00409-9, 2000.

**6. T. Cibic:** Discussion

What I miss here is a discussion of the ecological implications of these results at the lagoon mesoscale. In particular, a detailed discussion of the trophic state of the lagoon and a comparison with other Mediterranean lagoons.

**The authors:** We agree with this comment. In accordance, we added some sentences discussing about the ecological implications of our results at the lagoon mesoscale in the Discussion section (L474). Please also note that upon this addition, several references were added to the Reference list accordingly (L534, L566, L621, L627).

L474: "The results of the present experiment suggest that the climate-change related intensification of terrestrial runoffs could temporarily alter metabolic and trophic indexes of the water column of the lagoon during productive seasons (Trombetta *et al.* 2019), potentially shifting it toward heterotrophy and disrupting its trophic balance. Coupled with terrestrial runoff-induced shifts of microbenthic net community production toward heterotrophy (Liess *et al.* 2015), these alterations could interact with ongoing shifts occurring in the lagoon, such as the changes in trophic functioning toward mixotrophy and heterotrophy related to oligotrophication (Derolez *et al.* 2020b). Such consequences may also be seen in other Mediterranean lagoons, as turbidity and extreme flood events were reported to control phytoplankton abundance and phenology in oligotrophic Mediterranean coastal lagoons in Southern France and Corsica (Bec *et al.* 2011, Ligorini *et al.* 2022)."

L534: Bec, B., Collos, Y., Souchu, P., Vaquer, A., Lautier, J., Fiandrino, A., Benau, L., Orsoni, V., and Laugier, T.: Distribution of picophytoplankton and nanophytoplankton along an anthropogenic eutrophication gradient in French Mediterranean coastal lagoons. *Aquat. Microb. Ecol.* **63**:29-45. https://doi.org/10.3354/ame01480, 2011.

L566: Derolez, V., Malet, N., Fiandrino, A., Lagarde, F., Richard, M., Ouisse, V., Bec, B., and Aliaume, C.: Fifty years of ecological changes: Regime shifts and drivers in a coastal Mediterranean lagoon during oligotrophication. *Sci. Tot. Env.* **732**:139292. https://doi.org/10.1016/j.scitotenv.2020.139292, 2020b.

L621: Liess, A, Faithfull, C., Reichstein, B., et al.: Terrestrial runoff may reduce microbenthic net community productivity by increasing turbidity: a Mediterranean coastal lagoon mesocosm experiment. *Hydrobiologia* **753**:205-218. https://doi.org/10.1007/s10750-015-2207-3, 2015.

L627: Ligorini, V., Malet, N., Garrido, M., Derolez, V., Amand, M., Bec, B., Cecchi, P., and Pasqualini, V.: Phytoplankton dynamics and bloom events in oligotrophic Mediterranean lagoons: seasonal patterns but hazardous trends. *Hydrobiologia* **849**:2353-2375. https://doi.org/10.1007/s10750-022-04874-0, 2022.

---

## Author Comment (AC3)

Please note that modifications made in response to the comments of the Referee 1 (A. Regaudie-de-Gioux) are indicated on the document with the added parts highlighted in yellow and deleted parts , and modifications made in response to the comments of the Referee 2 (T. Cibic) are indicated on the document with the added parts highlighted in blue and deleted parts

[revised manuscript text omitted]

---

## Editor Decision (ED1)

Please note that modifications made in response to the comments of the Referee 1 (A. Regaudie-de-Gioux) are indicated on the document with the added parts highlighted in yellow and deleted parts crossed out in red, and modifications made in response to the comments of the Referee 2 (T. Cibic) are indicated on the document with the added parts highlighted in blue and deleted parts crossed out in purple.

[revised manuscript text omitted]

---

## Author Response (AR2)

Dear associate editor, thank you for your helpful comments and corrections. Please find below how we considered all your comments and corrections. The added parts are highlighted in green, while the removed parts are . Please note that most of the corrections were implemented directly on the manuscript.

**1.** L46. **Editor**: "I won't notice it further, please change it elsewhere in case you agree".

**Author's response**: According to your comment, we changed "R" to "CR" in the entire manuscript.

**2.** L59. **Editor**: "Please detail (in which way)?".

**Author's response**: According to your comment, we added precision to the sentence: "and  decrease the abundance of protozooplankton (Courboulès *et al.* 2023)."

**3.** L70. **Editor**: "Too vague, please detail".

**Author's response**: According to your comment, we added precision to the sentence: " Six mesocosms were used with…"

**4.** L87. **Editor**: "what does that mean? Needs to be much clearer".

**Author's response**: According to your comment, we added precision to the sentence: "The duration of the experiment was set  to 18 days  in order to monitor the responses of plankton at medium-term (multiple days to weeks), as interesting dynamics were already reported in control treatments during  previous experiments in the Thau Lagoon up to almost 3 weeks after the start of the experiment (Courboulès *et al.* 2021, Soulié *et al.* 2022a) However, the duration of the experiment was limited by COVID-19 pandemics restrictions, preventing from conducting a longer experiment."

**5.** L87. **Editor**: "seems to be a rotor pump, do you have any references showing that it does not significantly disrupt communities (especially large phytoplankton cells and mesozooplankton".

**Author's response**: We did not find any reference specifically showing that the type of pump that we have used does not damage organisms. However, we did not observe any damage on the organisms when counting them and identifying them on the microscope, even for the more fragile organisms, such as ciliates. We therefore added this information in the manuscript.

"In each mesocosm, the water column was continuously homogenized with a pump (Rule, Model 360) immersed at a depth of 1 m, resulting in a turn-over rate of approximately 3.5 $d^{-1}$. Observations performed with a microscope

indicated that organisms (phytoplankton, zooplankton) did not seem to be damaged by the mixing procedure, even fragile organisms such as ciliates."

**6.** L135. **Editor**: "underwater where??? This is not clear. Perhaps in plastic tanks filled with water?".

**Author's response**: Yes, bottles were kept in an opaque water-filled plastic tank. According to your comment, we added this information to the manuscript.

"After at least 6 hr of fixation during which bottles were kept underwater and in the dark in opaque plastic tanks filled with freshwater at room temperature, the DO concentration in each bottle was measured with an automated Winkler titrator (Methrom Metrohm 916-Ti-Touch) using a potentiometric titration method"

**7.** L168. **Editor**: "CRM from Dickson are NOT made for Ph reference".

**Author's response**: We agree with your comment. The Dickson references were only used for TA analysis. Considering pH analysis, an afterward comparison with purified m-Cp was performed to make sure that no potential drift happens. We added this information in the manuscript. Please note that upon this revision, a reference was added to the reference list.

"Certified reference seawater for carbonate chemistry (provided by Prof A. G. Dickson, Scripps, California) was used for TA analysis and to check the stability of pH analysis during the experiment pH and TA analysis. After the experiment, the m-Cp used was checked against a purified m-Cp batch (Liu *et al.* 2011), showing a difference < 0.005 pH units."

**8.** L267. **Editor**: "Is it correct, error bars are showing minimal and maximal values? Just to make sure as this would be the correct approach when having less than 3 replicates".

**Author's response**: Yes, error bars are showing minimal and maximal values. We added this information in the figure captions.

**"Error bars represent the range of the observations (min and max values)."**

**9.** L371. **Editor**: "No mixing of water in this experiment? If so, worth mentioning it."

**Author's response**: We agree with your comment and added this information in the sentence.

"Nevertheless, the sinking of the added soil during the experiment performed by Deininger *et al.* (2016), despite the use of a mixing pump to limit sedimentation, might have rapidly lessen light attenuation, possibly releasing phytoplankton from the negative effect of light limitation."